# Effects of the menstrual cycle phase on anterior cruciate ligament neuromuscular and biomechanical injury risk surrogates in eumenorrheic and naturally menstruating women: A systematic review

**Thomas Dos'Santos**[1,2]*, **Georgina K. Stebbings**[1,2], **Christopher Morse**[1,2], **Medha Shashidharan**[1,2], **Katherine A. J. Daniels**[1,2], **Andy Sanderson**[1,2]

**1** Department of Sport and Exercise Sciences, Musculoskeletal Science and Sports Medicine Research Centre, Manchester Metropolitan University, Manchester, United Kingdom, **2** Manchester Institute of Sport, Manchester Metropolitan University, Manchester, United Kingdom

* t.dossantos@mmu.ac.uk

## Abstract

### Background

Eumenorrheic women experience cyclic variations in sex hormones attributed to the menstrual cycle (MC) which can impact anterior cruciate ligament (ACL) properties, knee laxity, and neuromuscular function. This systematic review aimed to examine the effects of the MC on ACL neuromuscular and biomechanical injury risk surrogates during dynamic tasks, to establish whether a particular MC phase predisposes women to greater ACL injury risk.

### Methods

PubMed, Medline, SPORTDiscus, and Web of Science were searched (May-July 2021) for studies that investigated the effects of the MC on ACL neuromuscular and biomechanical injury risk surrogates. Inclusion criteria were: 1) injury-free women (18–40 years); 2) verified MC phases via biochemical analysis and/or ovulation kits; 3) examined neuromuscular and/or biomechanical injury risk surrogates during dynamic tasks; 4) compared ≥1 outcome measure across ≥2 defined MC phases.

### Results

Seven of 418 articles were included. Four studies reported no significant differences in ACL injury risk surrogates between MC phases. Two studies showed evidence the mid-luteal phase may predispose women to greater risk of non-contact ACL injury. Three studies reported knee laxity fluctuated across the MC; two of which demonstrated MC attributed changes in knee laxity were associated with changes in knee joint loading (KJL). Study quality (Modified Downs and Black Checklist score: 7–9) and quality of evidence were low to very low (Grading of Recommendations Assessment Development and Evaluation: very low).

**Data Availability Statement:** The data within this study are secondary data and available through the relevant articles referenced throughout. This is a systematic review, and thus, there is no data to provide.

**Funding:** The author received no specific funding for this work.

**Competing interests:** The authors have declared that no competing interests exist.

## Conclusion

It is inconclusive whether a particular MC phase predisposes women to greater non-contact ACL injury risk based on neuromuscular and biomechanical surrogates. Practitioners should be cautious manipulating their physical preparation, injury mitigation, and screening practises based on current evidence. Although variable (i.e., magnitude and direction), MC attributed changes in knee laxity were associated with changes in potentially hazardous KJLs. Monitoring knee laxity could therefore be a viable strategy to infer possible ACL injury risk.

## 1. Introduction

Female athletes (18–40 years old) are ~3.5 times more likely to sustain an ACL injury compared to male athletes [1], depending on sporting population [2]. Despite recent advancements in sports medicine and technology, ACL injury rates in female athletes are not declining [3–6] which is problematic as female sport participation rates are increasing [7, 8]. Female ACL injury incidence rates would therefore be predicted to increase in future due to greater participant exposures. Although risk factors related to skeletal anatomy [9–13], biomechanical movement strategies [14–17], neuromuscular activation patterns [15, 18–20], and biopsychosocial factors (i.e., inequities in socioeconomic status and skill training provision) may partially contribute to the sex disparity in ACL injury [3, 4, 21–23], one specific physiological risk factor increasing in interest is the role of fluctuations in ovarian sex hormones on ACL injury risk attributed to the menstrual cycle (MC).

Eumenorrheic and naturally menstruating women of reproductive age experience variations in ovarian sex hormones during different phases of the MC which can influence physiological systems and function [24–29]. For example, based on a 28-day MC length, the early follicular days 1–5 (Phase 1) is associated with low oestrogen and progesterone, whereas the highest oestrogen to progesterone ratio is observed during late follicular days 6–12 (Phase 2). During ovulation (Phase 3; days 13–15), oestrogen is lower (medium concentration) than phase 2 but higher than phase 1 with low progesterone levels, whereas the mid-luteal days 20–23 (Phase 4; ~7 days post ovulation) contains the highest progesterone concentrations, with relatively high oestrogen levels (> phase 1 and 3 but < than 2) [25, 26, 30]. Due to the different concentrations of ovarian sex hormonal profiles throughout the MC, phases associated with increased oestrogen may impact soft tissue compliance [31, 32], influence collagen formation and the tensile properties and integrity of ligaments (i.e., mechanical load tolerance) [33–35], impacting ligamentous and knee laxity [33, 34, 36], and neuromuscular function [24, 27, 30, 37, 38], and thus potentially increasing ACL injury susceptibility [39, 40].

There is, however, mixed and conflicting evidence that a specific MC phase may predispose female athletes to greater risk of non-contact ACL injury [39–42]. Notably, previous research in this area is generally confounded by methodological and research design limitations. These include inconsistencies in MC verification (i.e., lack of biomechanical analysis or ovulation kits, thus potential inclusion of anovulatory or luteal phase deficient women) and definitions [25–27], non-homogenous group profiling (i.e., describing injury in both the follicular and ovulatory phases [preovulatory] without considering the distinct hormonal variations in the early and late parts of each), inclusion of hormonal contraception (HC) users (distinctly different hormonal profiles) and contact ACL injuries, and use of unreliable injury recall or questionnaires.

As neuroexcitation [38], neuromuscular function [24, 27, 30, 37, 38], and ligamentous and knee laxity [33, 34, 36] can fluctuate throughout the MC, as well as psychological and perceptions of perceived effort and intensity [27], MC hormonal perturbations are likely to affect neuromuscular activation and coordination patterns during high impact tasks [39]. These changes may impact neuromuscular control and movement quality, which may influence the generation of hazardous mechanical loads associated with non-contact ACL injury risk during jump-landing and change of direction (COD) tasks [39]. To improve ACL injury mitigation strategies, injury screening protocols, and physical preparation and management of female athletes, greater understanding of how hormonal, neuromuscular, and biomechanical factors interrelate and influence the execution and movement quality of jump-landing and COD tasks across different MC phases is needed. The aim of this systematic review was to examine the effects of the MC on ACL neuromuscular and biomechanical injury risk surrogates, during dynamic, high impact tasks in eumenorrheic and naturally menstruating women. A secondary aim was to highlight the limitations, considerations, and future directions for research to improve our understanding regarding the effect of the MC on ACL injury risk. It was hypothesised that differences in neuromuscular and biomechanical injury risk surrogates would be observed between MC phases in eumenorrheic and naturally menstruating women. If specific MC phases may have potentially heightened injury risk, the findings may assist in ACL injury mitigation strategies, injury screening protocols, and physical preparation and management of female athletes.

## 2. Methods

This review conforms to the PRISMA 2020 statement guidelines [43] (S1 Checklist). A review protocol was not pre-registered for this review; however, the review methods were established prior to conducting the review.

### 2.1 Study inclusion and exclusion criteria

Consideration of Population, Intervention, Comparator, Outcomes, and Study design (PICOS) was used to establish the parameters within which the review was conducted [44]. The PICOS strategy is presented in Table 1. Studies that did not meet the PICOS criteria were excluded from the review.

### 2.2 Search strategy

A literature search was performed using PubMed, Medline (OVID), SPORTDiscus, and Web of Science databases by two reviewers (TDS and MS) from May 2021 to July 2021 with the final search date of 2$^{nd}$ July 2021. Citation tracking on Google Scholar was also used to identify any additional material. A schematic of the search methodology in accordance with established guidelines [43] is presented in Fig 1.

Search terms were as follows:

1. ("Anterior cruciate ligament" OR "knee" OR "ACL")

2. ("biomechanic" OR "biomechanics" OR "biomechanical" OR "neuromuscular" OR "injury" OR "kinetic" OR "kinematic" OR "electromyography" OR "muscle activation" OR "biomec*" or electromy*")

3. ("menstrual phase" OR "menstrual cycle" OR "menstrual" OR "menstruation" OR "follicular phase" OR "luteal phase" OR "ovulation" OR "ovulatory")

4. 1 AND 2 AND 3

**Table 1. Consideration of population, intervention, comparator, outcomes and study design.**

| | |
|---|---|
| **Population** | Participants included women who were: a) aged between 18 and 40 years old; b) Healthy Eumenorrheic / naturally menstruating women of reproductive age (i.e., post-menarche and pre-menopausal) who experience ovulation confirmed via biochemical analysis or ovulation kits; c) free from any menstrual-related dysfunctions (e.g., amenorrhea) or any other conditions (e.g., pregnancy, eating disorders or disordered eating, low energy availability and relative energy deficiency syndrome) known to affect the hypothalamic–pituitary–ovarian axis [24]; d) never previously sustained a severe knee injury such as an ACL injury, and no previous lower-limb injury which required surgery or no previous lower limb injury within a year prior to testing; e) engaged in physical activity or sport from recreational to elite playing status [45]. Studies that contained participants with previous or current history of polycystic ovary syndrome (PCOS) were excluded. |
| **Intervention / Method** | A specific intervention was not investigated but participants were required to meet the population criteria above. Studies must have examined biomechanical and/or neuromuscular surrogates of non-contact ACL injury risk during a pre-planned or unplanned bilateral or unilateral landing / jump-landing, deceleration, or COD task using 3D motion and/or GRF analysis, 2D kinematic analysis, goniometry, and/or surface electromyography (sEMG). Only studies that verified participants' MC phase via biochemical analysis (i.e., blood/serum analysis) and / or ovulation kits were included, in line with McNulty et al. [24]. |
| **Comparator** | To determine the effect of MC phase on neuromuscular and biomechanical injury risk surrogates, included studies must compare an outcome measure (i.e., surrogate of ACL injury risk) at a minimum of 2 phases of MC in line with McNulty et al. [24]. Comparisons must have been made between two of the following MC phases in line with classifications provided in previous research [24, 25]: early follicular (days 1–5), late follicular (days 6–12), ovulation (days 13–15), early luteal (days 16–19), mid-luteal (days 20–23) and late luteal (days 24–28). |
| **Outcomes** | Studies must have included precise mean and SD or SEs for injury risk surrogates for all phases examined. Subsequently, a change in injury risk surrogate between MC phases denotes a potential change in non-contact ACL injury risk. Biomechanical non-contact ACL injury risk surrogates (i.e., factor which has been shown to increase ACL loading / strain / KJL or prospectively shown to predict injury) included [9, 46, 47]: knee abduction, rotation, flexion moments / impulse (knee joint loads) due to the propensity to increase ACL strain [47, 48], or studies that investigated ACL loading via musculoskeletal modelling [49, 50] or measured proximal anterior tibial shear [51]. Technical, kinetic, or kinematic determinants of surrogates of injury risk (i.e., KJL) [9, 46, 52] related to quadriceps, ligament, trunk, and/ or leg dominance, such as: vertical / posterior GRF / impulse; initial or peak lateral trunk flexion/rotation angle, hip internal rotation angle, knee valgus / internal rotation angle, knee flexion / hip flexion angle, and foot progression angles were also considered as biomechanical injury risk surrogates. Because neuromuscular activation patterns have been shown to predict non-contact ACL injury [53, 54], and has the potential to support ligament unloading [55], or increase ACL loading [56–59]; studies that examined outcome measures related to quadriceps femoris, hamstrings, gluteal groups, gastrocnemius, and / or soleus preactivation, time to activation, peak activation at pre ground contact, initial contact and / or over weight-acceptance were included and considered as injury risk surrogates. |
| **Study design** | Clinical studies were included for analysis if they: a) were a peer reviewed full published article in English using human participants; b) had the primary or secondary objective of monitoring changes in ACL injury risk surrogates between MC phases; and c) included within-group comparison for phases of MC with outcome measures clearly taken during two or more MC phases. |
| **Other data extraction** | The following data was extracted and recorded in a spreadsheet:<br>1. author names, publication year<br><br>2. sample size and participant characteristics including sport(s), playing level/status, training history, strength training history/status/profile<br><br>3. timing of injury risk surrogate (outcome measures) during the MC and whether testing order was randomised<br><br>4. how MC phase was identified and verified (method)<br><br>5. how ACL injury risk surrogate was assessed (methods)<br><br>6. reliability and familiarisation stated for outcome measures / tasks (if reported)<br><br>7. outcome measures with results |

**Key:** ACL: Anterior cruciate ligament; MC: Menstrual cycle; KJL: Knee joint load; COD: Change of direction; 2D: Two-dimensional; 3D: Three-dimensional; GRF: Ground reaction force

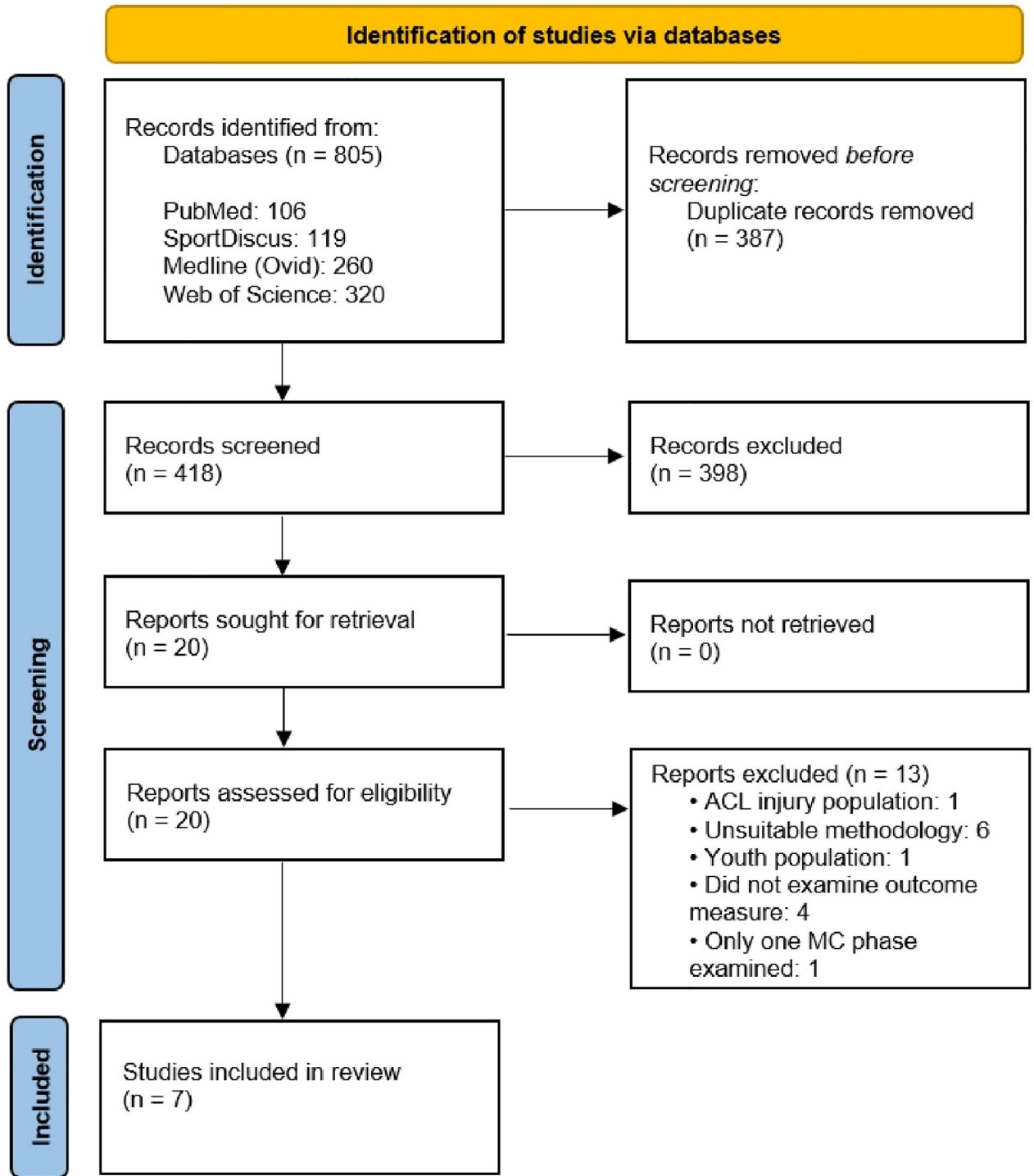

**Fig 1. Flow chart illustrating the different phases of the search strategy in accordance with PRISMA guidelines.** Adapted from Page et al. [43].

Subsequently, bibliographies of prospectively eligible studies were compiled and hand searched by two independent reviewers to screen for further suitable studies. Studies were first assessed based on title and abstract to identify potentially eligible studies, the full text of these studies was then read to confirm if they met the eligibility criteria. If disagreement on eligibility

occurred between the two reviewers (TDS and MS), a third independent reviewer (AS) was consulted, and their decision deemed as final.

## 2.3 Quality assessment of included studies and quality of evidence

Study quality was assessed by two reviewers (TDS and MS) and independently verified by one reviewer (AS). The appraisal tool was based on the Downs and Black checklist for measuring study quality and was modified according to McNulty et al. [24] who conducted a related review examining the effect of MC on exercise performance, and thus developed a more appropriate tool for the present review. The modified Downs and Black checklist comprised 15 outcomes, from five domains: (1) reporting; (2) external validity; (3) internal validity—bias; (4) internal validity—confounding; and (5) power. A maximum attainable score of 16 could be awarded, whereby study quality was categorised as follows: "high" (14–16); "moderate" (10–13); "low" (6–9); or "very low" (0–5) [24]. The results of the Downs and Black assessment were used to assign an *a priori* quality rating to each study. In accordance with McNulty et al. [24], the *a priori* rating was then either maintained, or downgraded a level, based on the response to two questions that were considered key to the directness of these research studies: Q.1) was the MC phase confirmed using blood samples? If the authors reported using blood samples to confirm MC phase, the *a priori* rating was maintained and if not, the study was downgraded a level (e.g., a study that started out as "high" in quality, but did not confirm MC phase using a blood sample, drops to "moderate" in quality); and Q.2) was the MC phase confirmed using urinary ovulation detection kits? If the authors reported the use of a urinary ovulation detection kit to identify MC phase, the Q.1 rating was maintained; if not, the study was downgraded a level. As such, the maximum rating for any study that does not use serum analysis or urinary ovulation detection kits to identify and verify MC phase is "low" [24].

Finally, the Grading of Recommendations Assessment, Development, and Evaluation (GRADE) approach was implemented to further assess the quality of evidence obtained from the present review [60]. The tool was applied for surrogates of ACL injury risk (Table 1; outcomes) for five determinants: risk of bias, inconsistency, imprecision, indirectness, and publication bias [61]. As all studies in this review were experimental / clinical studies (i.e., repeated measures observations with no formal intervention or treatment), the GRADE scores initially start with a low rating. The overall quality of evidence within the studies was upgraded for factors such as large effect sizes or dose-response relationships (i.e., directness) or downgraded for factors including risk of bias (i.e., control of confounding factors), imprecision (i.e., reporting of confidence intervals and *p* values), inconsistency (i.e., reported results / effects), indirectness (i.e., outcome measures and comparisons between MC phases) and publication bias [60, 61].

## 2.4 Data collation

Quantitative data pertaining to study methodology, participant characteristics, MC phase and verification, ACL injury risk surrogates, reliability measures, and results (Table 1) were obtained for qualitative analysis by two independent reviewers (TDS and MS). Results were collated through identifying significant ($p < 0.05$) and non-significant findings ($p > 0.05$) for outcome measures and correlational R values where applicable, while percentage changes and effect sizes were also extracted if provided by the authors.

## 3. Results

### 3.1 Literature search

Fig 1 illustrates a flow chart which summarises the results of the systematic search process.

## 3.2 Study characteristics and findings

Seven studies met the inclusion criteria for this review [33–35, 62–65]. An overview of the methodological quality assessment is provided in Table 2, while study characteristics and findings are presented in Tables 3 and 4. Sample sizes ranged from 10 to 71 [33–35, 62, 64, 65], with Shultz et al. [63] examining a substantially larger sample size of 71 compared to the next highest (*n* = 28). Participant characteristics were generally poorly described across all studies with no study providing any information pertaining to specific sports, skill level, or resistance training history.

Table 3 provides specific information pertaining to tasks examined (primarily bilateral and unilateral jumping), biomechanical and neuromuscular measures (primarily via 3D analysis *n* = 6 and EMG *n* = 3), injury risk surrogates, and measurement techniques for the included studies. Additionally, Table 3 provides information pertaining to no hormonal contraception usage (*n* = 4), MC phase verification method (i.e., urinary luteal hormone measurement *n* = 5 or blood samples *n* = 5), the MC phases examined (ranging from 2–5 phases in accordance with recommendations [24]) for the included studies.

With respect to the effect of MC phase on neuromuscular and biomechanical non-contact ACL injury risk surrogates, four studies observed no significant or meaningful differences between MC phases [33, 34, 62, 65], while two studies [35, 64] showed evidence that the mid-luteal phase may predispose women to greater risk of non-contact ACL injury compared to the early and late follicular phase based on neuromuscular activity and landing kinematics. Two studies [33, 34] showed that knee laxity fluctuated throughout the MC phases and the change in knee laxity was associated with changes in KJLs; however, considerable individual variation was observed with respect to the MC phase which elicited the greatest knee laxity and KJL. One study [63] also showed that increases in sagittal and frontal knee laxity were associated with increases in knee valgus motion, but muscle activity and KJLs were not significantly different. The researchers [63] examined landing mechanics during the periods of lowest and maximum laxity during the early follicular and mid-luteal phases but the authors do not clearly specify which phase elicited the highest or lowest laxity.

Only one study randomised the testing order across MC phases [64], with three studies confirming that testing order was non-randomised [33, 34, 62]. The remaining studies did not verify if the testing order was randomised [35, 63, 65]. Only one study clearly stated that testing conditions were standardised between MC phases [65], while one study confirmed that the examiner was blinded to MC phase [35]. Only two studies stated that a familiarisation period or session was provided prior to data collection [62, 63]. No study provided any reliability measures pertaining to the neuromuscular or biomechanical outcome measures [33–35, 62–65], nor did any study compare and interpret the change in outcome measure in relation to measurement error [33–35, 62–65]. No study examined joint-joint coordination changes between MC phases [33–35, 62–65], and only one study included a form of temporal analysis [63] while the remaining studies generally conducted discrete point analysis [33–35, 62, 64, 65].

## 3.3 Assessment of methodological quality and quality of evidence

Methodological quality assessment data is presented in Table 2. Three [33–35] and four [62–65] studies were classed as low and very low, respectively, with scores ranging from 7–9. All studies were provisionally (*a priori*) scored as low; however, two studies were downgraded due to the failure to confirm MC phases using blood samples [62, 63], with two more studies downgraded for not verifying MC phases using ovulation kits [64, 65] in accordance with McNulty et al. [24]. Thus, only three studies [33–35] maintained their *a priori* quality rating.

**Table 2. Methodological quality assessment of included studies and quality of evidence.**

| | Study | 1 | 2 | 3 | 4 | 5 | 6 | 7 |
|---|---|---|---|---|---|---|---|---|
| | Author | Abt et al. (2007) | Chaudhari et al. (2007) | Dedrick et al. (2008) | Okazaki et al. (2017) | Park et al. (2009) AMJSM | Park et al. (2009) BJSM | Shultz et al. (2012) |
| Q | Title | | | | | | | |
| | *Reporting* | | | | | | | |
| 1 | Is the hypothesis/aim/objective of the study clearly described? Yes = 1 No = 0 | 1 | 1 | 1 | 1 | 1 | 1 | 1 |
| 2 | Are the main outcomes to be measured clearly described in the introduction or methods section? If the main outcomes are first mentioned in the results section, answer no. Yes = 1 No = 0 | 1 | 1 | 1 | 1 | 1 | 1 | 1 |
| 3 | Are the characteristics of the participants included in the study clearly described? In observational studies, inclusion and/or exclusion criteria should be given. In case-control studies, inclusion and/or exclusion and the source of controls should be given. Yes = 1 No = 0 | 0 | 0 | 0 | 0 | 0 | 0 | 0 |
| 4 | Were the tested menstrual cycle phases clearly described? Answer yes if the precise criteria used to define phase were provided, answer no if the exact phase tested cannot be ascertained (*e.g.*, vague language such as "early" or "late" were used, without defining the criteria) Yes = 1 No = 0 | 1 | 0 | 1 | 1 | 1 | 1 | 1 |
| 5 | Are the main findings of the study clearly described? Simple outcome data should be reported for all major findings so the reader can check the major analyses and conclusions. This does not cover statistical tests which are addressed in other questions. Yes = 1 No = 0 | 1 | 1 | 1 | 1 | 1 | 1 | 1 |
| 6 | Does the study provide estimates of the random variability in the data for the main outcomes? In non-normal data, inter-quartile range should be reported. In normal data, standard deviation, standard error or confidence intervals should be reported. Yes = 1 No = 0 | 1 | 1 | 1 | 1 | 1 | 1 | 1 |
| | *External validity* | | | | | | | |
| 7 | Were the subjects confirmed as non-hormonal contraceptive users, for at least three months prior to participation? Yes = 1 No = 0 Unable to determine = 0 | 0 | 0 | 1 | 0 | 1 | 1 | 1 |
| | *Internal validity–bias* | | | | | | | |
| 8 | Was at least one familiarization trial conducted prior to exercise testing? Yes = 1 No = 0 Unable to determine = 0 | 1 | 0 | 0 | 0 | 0 | 0 | 1 |
| 9 | Were the exercise test conditions adequately standardised (taking into consideration factors including time of day, prior nutritional intake [including caffeine] and prior exercise). Yes (all relevant factors standardised) = 2 Yes (some relevant factors standardised) = 1 Exercise testing unstandardized = 0 Unable to determine = 0 | 0 | 2 | 0 | 0 | 0 | 0 | 0 |
| 10 | If any of the results of the study were based on 'data dredging' was this made clear? Any analyses that had not been planned at the outset should be clearly indicated. If no retrospective subgroup analyses were reported, then answer yes. Yes = 1 No = 0 Unable to determine = 0 | 0 | 0 | 0 | 0 | 0 | 0 | 0 |
| 11 | Were statistical tests used to assess the main outcomes appropriate? The statistical techniques used must be appropriate to the data and the research question. Yes = 1 No = 0 Unable to determine = 0 | 1 | 1 | 1 | 1 | 1 | 1 | 1 |

(*Continued*)

**Table 2.** (Continued)

| Q | Title | 1 Abt et al. (2007) | 2 Chaudhari et al. (2007) | 3 Dedrick et al. (2008) | 4 Okazaki et al. (2017) | 5 Park et al. (2009) AMJSM | 6 Park et al. (2009) BJSM | 7 Shultz et al. (2012) |
|---|-------|------|------|------|------|------|------|------|
| | **Study** | | | | | | | |
| | **Author** | Abt et al. (2007) | Chaudhari et al. (2007) | Dedrick et al. (2008) | Okazaki et al. (2017) | Park et al. (2009) AMJSM | Park et al. (2009) BJSM | Shultz et al. (2012) |
| | *Reporting* | | | | | | | |
| 12 | Were the main outcome measures used accurate (*i.e.*, valid and reproducible)? For studies where the validity and reproducibility of outcome measures are clearly described, the question should be answered yes. For studies which refer to other work that demonstrates the outcome measures are accurate, answer yes. Yes = 1 No = 0 Unable to determine = 0 | 0 | 0 | 0 | 0 | 0 | 0 | 0 |
| | *Internal validity–confounding (selection bias)* | | | | | | | |
| 13 | Was the order of phase testing randomised? Yes = 1 No = 0 Unable to determine = 0 | 0 | 0 | 1 | 0 | 0 | 0 | 0 |
| | *Power* | | | | | | | |
| 14 | Did the study have sufficient power to detect an *a priori* specified scientifically important effect at a pre-determined probability threshold? Answer yes if they included a power calculation, and no if not. Yes = 1 No = 0 | 0 | 0 | 0 | 0 | 0 | 0 | 0 |
| 15 | Was study retention > 85%? Yes = 1 No = 0 Unable to determine = 0 | 1 | 1 | 1 | 1 | 1 | 1 | 1 |
| | **Grading (assign an *a priori* study quality rating based on the modified Downs and Black checklist, so all studies will start out as being of "high", "moderate", "low", "very low")** | | | | | | | |
| 16 | Identify if menstrual cycle phase was confirmed using blood samples. If yes, the *a priori* rating is maintained and this is the final study quality rating. If not, the study is downgraded a level (*e.g.*, a study that started out as high, drops to moderate). | No | Yes | Yes | Yes | Yes | Yes | No |
| 17 | Identify if menstrual cycle phase was confirmed using ovulation kits. If yes, the Q1. rating is maintained. If no, the study is downgraded another level (*e.g.*, a study that started out high, drops to low). This means that the maximum rating that any study that does not use blood analysis or ovulation kits is "low" or "very low". | Yes | No | No | Yes | Yes | Yes | Yes |
| | **Rating** | 8 | 7 | 9 | 7 | 8 | 8 | 9 |
| | **Downs and Black checklist (maximum score attainable = 16). Study quality was categorised as follows: "high": 14–16; "moderate": 10–13; "low": 6–9; "very low": 0–5)** | | | | | | | |
| | *A priori* | Low | Low | Low | Low | Low | Low | Low |
| | **Final classification** | Very low | Very low | Very low | Low | Low | Low | Very low |

GRADE: Risk of bias: high (-); Inconsistency; yes (-); Imprecision: yes (-); Indirectness: no (+); Publication bias (-); moderate (-). Level of Evidence: Very low

Key: GRADE: Grading of Recommendations Assessment, Development and Evaluation; Q: Question.

There was a small number of studies included which were heterogenous in methodology precluding a meta-analysis; however, we assessed overall quality of evidence using GRADE, and found it to be very low (Table 2).

## 4. Discussion

The primary finding from this systematic review is that it is inconclusive whether a particular MC phase predisposes eumenorrheic and naturally menstruating women to greater non-contact ACL injury risk. Mixed findings from seven studies regarding the effects of the MC on ACL neuromuscular and biomechanical injury risk surrogates were observed, with very low quality of evidence. Four studies reported no meaningful differences in neuromuscular or biomechanical ACL injury risk surrogates between MC phases [33, 34, 62, 65], while two studies

**Table 3. Synthesis of methodologies and research designs of eligible studies.**

| Study | Author | MC verification | | MC phases examined | | | | | | Task | | | |
|---|---|---|---|---|---|---|---|---|---|---|---|---|---|
| | | Blood / Biochemical | Ovulation kits | Early follicular | Late follicular | Ovulatory | Mid-luteal | Follicular | Luteal | Cut | Jump-stop | Drop jump | Drop-Landing |
| 1 | Abt et al. (2007) | ☒ | ✓ | ✓ | ☒ | ✓ | ✓ | ☒ | ☒ | ☒ | ✓ | ☒ | ☒ |
| 2 | Chaudhari et al. (2007) | ✓ | ☒ | ☒ | ☒ | ✓ | ☒ | ✓ | ✓ | ☒ | ✓ | ✓ | ✓ |
| 3 | Dedrick et al. (2008) | ✓ | ☒ | ✓ | ✓ | ☒ | ✓ | ☒ | ☒ | ☒ | ☒ | ✓ | ☒ |
| 4 | Okazaki et al. (2017) | ✓ | ✓ | ✓ (spans late follicular) | ✓ | ✓ (spans late follicular) | ✓ | ☒ | ☒ | ☒ | ☒ | ☒ | ✓ |
| 5 | Park et al. (2009) AMJSM | ✓ | ✓ | ✓ (spans late follicular) | ☒ | ✓ | ✓ | ☒ | ☒ | ✓ | ✓ | ☒ | ☒ |
| 6 | Park et al. (2009) BJSM | ✓ | ✓ | ✓ (spans late follicular) | ☒ | ✓ | ✓ | ☒ | ☒ | ✓ | ☒ | ☒ | ☒ |
| 7 | Shultz et al. (2012) | ☒ | ✓ | ✓ | ☒ | ☒ | ✓ | ☒ | ☒ | ☒ | ☒ | ✓ | ☒ |

| Study | Author | Methods / Outcome measures | | | | | Data collection / research design | | | | Population | | |
|---|---|---|---|---|---|---|---|---|---|---|---|---|---|
| | | Moments | GRF | Kinematics | sEMG | Laxity | Familiarisation (F) or reliability (R) | Testing order randomised | Assessor blinding | Testing standardisation | Population clearly stated | MC length clearly stated | Non-HC usage > 2 months |
| 1 | Abt et al. (2007) | ✓ | ✓ | ✓ | ☒ | ☒ | ✓ (F) | ☒ | ☒ | ☒ | ☒ | ☒ | ☒ |
| 2 | Chaudhari et al. (2007) | ✓ | ✓ | ✓ | ☒ | ☒ | ☒ | ☒ | ☒ | ✓ | ☒ | ✓ | ☒ |
| 3 | Dedrick et al. (2008) | ☒ | ☒ | ✓ | ✓ | ☒ | ☒ | ✓ | ☒ | ☒ | ☒ | ✓ | ✓ |
| 4 | Okazaki et al. (2017) | ✓ | ✓ | ✓ | ✓ | ☒ | ☒ | ☒ | ✓ | ☒ | ☒ | ☒ | ☒ |
| 5 | Park et al. (2009) AMJSM | ✓ | ✓ | ✓ | ☒ | ✓ | ☒ | ☒ | ☒ | ☒ | ☒ | ✓ | ✓ |
| 6 | Park et al. (2009) BJSM | ✓ | ✓ | ✓ | ☒ | ✓ | ☒ | ☒ | ☒ | ☒ | ☒ | ✓ | ✓ |
| 7 | Shultz et al. (2012) | ✓ | ✓ | ✓ | ✓ | ✓ | ✓ (F) | ☒ | ☒ | ☒ | ☒ | ✓ | ✓ |

**Key:** MC: Menstrual cycle; HC: Hormonal contraception; GRF: Ground reaction force; sEMG: Surface electromyography; F: Familiarisation; R: Reliability

**Table 4. Summary of literature that has investigated the effects of MC phase on neuromuscular and biomechanical non-contact ACL injury risk surrogates.**

| Author | Sample size n = Population details (i.e., sport, playing level, age, height) Population training, strength, skill history | Population menstrual / health status and length MC phase method verification and details | Dynamic task and methods Key outcome measures / variables Reliability / familiarisation reported (y/n) | Results and implications (i.e., does a certain MC phase predispose athlete to greater risk of injury? | Comments |
|---|---|---|---|---|---|
| Abt et al. (2007) | n = 10 Age: 21.4 ± 1.4 years; height: 1.67 ± 0.06 m; mass: 59.9 ± 7.4 kg Physically active women recruited from a local university. Specific sport, skill, or strength training history not stated. | Eumenorrheic and ovulating. MC length and frequency per year not stated. Phases were defined based on first day of menses. Urine samples collected daily starting on day 10 until a positive test was received using ovulation detection kit. Positive test indicated start of post ovulatory phase. Mid luteal phase was defined as starting 7 days after a positive ovulation test (21–23 of MC). Ovulation was also confirmed by screening mid-luteal progesterone threshold level. Testing occurred three times; on day 3 of menses (early follicular), 24–36 hours post-positive ovulation test, mid-luteal (21–23) / 7 days post ovulation detection. | Single leg stop-jump task - 3D motion and GRF analysis Knee flexion and valgus angle, peak proximal tibial anterior shear force. Knee flexion and valgus moment at peak anterior tibial shear force. From initial contact to peak to peak knee flexion / valgus. One month familiarisation period. No reliability measures reported for outcome measures. | • Estradiol ↑ during post-ovulatory and mid-luteal vs early follicular <br>• Progesterone ↓ in early follicular and post-ovulatory <br>• ↔ in knee flexion excursion, knee valgus excursion, peak proximal tibial anterior shear force, flexion moment at peak proximal tibial anterior shear force, or valgus moment at peak proximal tibial anterior shear force between MC phases. <br>• No difference in knee biomechanical injury risk surrogate were found in the phases. This implies that there may not be a greater risk of ACL injury associated with one particular MC phase. | Specific sport, skill, or strength training history not stated. 2 phases of MC verified. Limited number of biomechanical variables in relation to ACL examined; only knee kinetic and kinematics; ACL injury multi-segment mechanism (i.e., hip, trunk and foot). Small sample size. MC length not stated. Testing order was non-randomised. One subject removed for anovulatory cycle. No individual analysis which may conceal differences. No reliability measures reported for outcome measures. No sEMG. No temporal analysis / coordination examined. MC phases not verified with blood samples. HC usage not confirmed. Changes were not examined relative to measurement error |
| Chaudhari et al. (2007) | n = 12 Age: 19.1 ± 1.0 years; height: 1.7 ± 0.1 m; weight: 595 ± 98 N Women who participated in recreational sports at a local university. Specific sport, skill, or strength training history not stated. | Verified MC length for 2 cycles. Women were tested twice for each phase of the MC (follicular, luteal, ovulatory), as determined from serum analysis. Onset of the ovulatory phase of the cycle was determined using the PC-2000 saliva testing system (I.M. P., Manschau, Germany), which gives a positive result 3 to 5 days before ovulation. The subjects used this system once a day for 60 days before testing. Onset of luteal phase (~day 15) was estimated from saliva testing data, and a serum blood test for estradiol, progesterone, and luteinizing hormone was performed. Specific dates of the 3 phases not provided. Testing conditions standardised between phases. | Bilateral horizontal jump, bilateral vertical jump, and unilateral drop from a 30-cm box on the left leg. Limited 3D motion and GRF analysis—5 marker link model Lower-limb kinematics (at foot strike and max knee flexion) and peak externally applied moments were calculated (hip adduction moment, hip internal rotation moment, knee flexion moment, knee abduction moment) during weight acceptance. No familiarisation or reliability measures reported for outcome measures. | • ↔ in knee moments or knee angle between MC phases <br>• No difference in knee biomechanical injury risk surrogate were found in the phases. This implies that there may not be a greater risk of ACL injury associated with one particular MC phase. | Specific sport, skill, or strength training history not stated. Small sample size. MC phases not clearly described. Unclear if non-randomised testing order was used. No individual analysis which may conceal differences. Large variability in mean/SD. Limited marker model used for 3D motion analysis. No reliability measures reported for outcome measures. No sEMG. No temporal analysis / coordination examined. MC phases not verified with ovulation kit. HC usage not confirmed. Changes were not examined relative to measurement error. |

*(Continued)*

**Table 4.** (Continued)

| Author | Sample size<br>n = Population details<br>(i.e., sport, playing level,<br>age, height)<br>Population training,<br>strength, skill history | Population menstrual / health<br>status and length<br>MC phase method verification<br>and details | Dynamic task and methods<br>Key outcome measures /<br>variables<br>Reliability / familiarisation<br>reported (y/n) | Results and implications (i.e.,<br>does a certain MC phase<br>predispose athlete to greater risk<br>of injury? | Comments |
|---|---|---|---|---|---|
| Dedrick et al. (2008) | n = 22<br>Age: 20.5 ± 1.9 years; height: 164.9 ± 5.6 cm; mass: 62.1 ± 13.7 kg; and Wojtys et al. [66] activity scale 5.4 ± 1.4 hr/week Recreationally active women with no formal history of jump-landing training.<br>Specific sport, skill, or strength training history not stated. | Participants had regular menses for at least one year. Eumenorrhea was defined as cycles occurring at regular intervals ranging from 26 to 32 days. Cycles established for 2 months prior to testing.<br>MC length 28.3 ± 1.5 days. Participants non-HC usage ≥ 3 months prior to testing. Blood assays verified sex hormone levels and cycle phase. Random assignment of participants to testing time: 11 subjects beginning data collection in the early follicular phase (day 1–3), 8 in mid-luteal phase (day 21–24), and 7 in the late follicular phase (day 11–13) of the menstrual cycle. | Bilateral drop jumps from 0.5m.<br>sEMG of six muscles and goniometer data for knee angles. Varus/valgus knee angle and sEMG activity from six lower extremity muscles (GM, ST, vastus lateralis, vastus medialis oblique, tibialis anterior, and the lateral head of the gastrocnemius). Angle data between initial contact and 500ms of ground contact. No familiarisation or reliability measures reported for outcome measures. | • ↔ knee valgus angle or time to reach knee valgus angle between MC phases<br>• ST muscle exhibited onset delays relative to ground contact during mid-luteal phase, and demonstrated a significant difference in onset time between early and late follicular phases.<br>• Muscle timing differences between the GM and ST were decreased in the mid-luteal phases compared to early follicular phases influencing co-activation.<br>• ↔ temporal muscle sequencing / sequential recruitment between MC phases<br>• MC phases appears not to influence knee valgus angle; however, may alter muscle activity (onset and timing) particularly ST and activation of GM and ST during mid-luteal phase which may increase ACL risk. | Participants excluded if history of formalized jump training. Specific sport, skill, or strength training history not stated. Random assignment of cycles which is positive. Only limited number of variables examined. Only one task examined. No reliability measures reported for outcome measures. Limited number of biomechanical variables in relation to ACL injury risk examined; only knee kinematics; ACL injury multi-segment mechanism (i.e., hip, trunk and foot). No temporal analysis / coordination examined. MC phases not verified with ovulation kit. Changes were not examined relative to measurement error. |
| Okazaki et al. (2017) | n = 28<br>Age: 21.0 ± 0.8 years; height: 158.1 ± 6.0 cm; mass: 52.8 ± 6.5 kg<br>Specific sport, skill, activity, or strength training history not stated. | Healthy but MC length not stated<br>Blood sample verification menstrual phase (1st-5th day; actually, early follicular), follicular phase (7th-10th day; actually, late follicular), ovulation phase (12th-15th day), and luteal phase (7th-9th day from ovulation; actually, mid-luteal).<br>MC cycles verified over 3 cycle history. MC phases verified with assay of serum hormonal levels. The test examiner was blinded to the MC of the participants. Ovulation kit used to detect ovulation. | Unilateral drop-landing 0.30 m. 3D motion and GRF analysis and sEMG<br>Peak GRF and time to peak GRF. Hip, knee, and ankle kinematics. EMG activity 150 ms before activity. GM, ST, BF, and RF.<br>No familiarisation or reliability measures reported for outcome measures. | Mid-luteal phase:<br>• Time to peak GRF ↓ vs. early follicular phase (6%)<br>• Hip internal rotation and knee valgus ↑ vs early follicular (43% and 34%, respectively)<br>• Knee flexion ↓ vs early and late follicular phases (7–9%)<br>• Ankle dorsi-flexion ↓ vs late follicular phase (11%)<br>• Ankle adduction and eversion ↑ vs early and late follicular phases (26–46%, and 27–33%, respectively)<br>• GM ↓ activation before landing vs early and late follicular phases (20–22%)<br>• ↔ between MC phases for BF, ST, or RF muscle activity<br>• Mid-luteal phase may predispose women to greater risk of non-contact ACL injury based on changes in biomechanical and NMS injury risk surrogates | Incorrect definition of MC phases, specifically stating menstrual phases. Specific sport, skill, or strength training history not stated. The examiner was blinded which is positive but unclear if testing order was randomised. No knee joint loads were examined. No reliability measures reported for outcome measures. No individual analysis which may conceal differences. No temporal analysis / coordination examined. HC usage not confirmed. Changes were not examined relative to measurement error. |

**Table 4.** (Continued)

| Author | Sample size<br>n = Population details<br>(i.e., sport, playing level,<br>age, height)<br>Population training,<br>strength, skill history | Population menstrual / health<br>status and length<br>MC phase method verification<br>and details | Dynamic task and methods<br>Key outcome measures /<br>variables<br>Reliability / familiarisation<br>reported (y/n) | Results and implications (i.e.,<br>does a certain MC phase<br>predispose athlete to greater risk<br>of injury? | Comments |
|---|---|---|---|---|---|
| Park et al. (2009) AMJSM | n = 26<br>Age: 22.7 ± 3.3 years; height: 170.1 ± 7.1 cm; mass: 65.0 ± 9.6 kg<br>Recreationally active women who played Sports—activity levels 8.7 ± 4.6 hr/week<br>Specific sport, skill, or strength history not stated. | MC length: 28.9 ± 2.7 days Follicular (Early) (3–7), ovulation (24–48 hours after detection of oestrogen surge—ovulation kit) and luteal (mid-luteal) (~7 days post ovulation) phases—serum hormone. Non-randomised order. Blood samples collected at 3 time points to verify.<br>Timing of the third test varied and ranged between 19 and 26 days (22.85 ± 3.22 days) depending on the length of the participants' MC. Participants non-HC usages ≥ 3 months. | 45˚ pre-planned cut (CUT45); target speed 3.5 m/s. 3D motion and GRF analysis Bilateral jump and stop action—self-selected speed and kept consistent between sessions. 3D motion and GRF analysis Knee laxity—KT-2000 arthrometer 89N and compared 3D data between participants with low, medium, and high laxity Knee laxity (mm), peak knee angle, and knee joint moment (Nm) and knee joint impulse (Nms). Internal knee adduction, knee flexion, and int/ext rotation moment during stance phase. Reliability measures reported for knee laxity only. | • ↔ for any parameter for both tasks between MC phases<br>• Of the 26 participants, 13 showed the lowest knee laxity during the early follicular phase; 3 were lowest during ovulation, and 10 lowest during mid-luteal phase. 14 of 26 participants displayed highest knee laxity during ovulation compared with 10 participants during the early follicular phase and only 2 participants during the mid-luteal phase.<br>• ↑ CUT45 knee adduction (valgus) impulse (0.31 ± 0.30 Nms) in high vs low knee laxity group (0.22 ± 0.21 Nms).<br>• ↑ jump-stop knee adduction moment (49.77 ± 23.05 Nm) in high vs medium knee laxity group (40.23 ± 20.42 N·m).<br>• ↑ jump-stop knee external rotation moment and impulse in high vs low knee laxity group (external rotation impulse: 0.47 ± 0.42 Nms vs 0.27 ± 0.27 Nms; external rotation moment: 8.28 ± 4.45 N·m vs 6.04 ± 3.51 N·m) during jumping and stopping.<br>• 1.3-mm difference in knee laxity ↑ ~30% in knee adduction impulse during CUT45, ↑ ~20% in knee adduction moment, and ↑ ~ 20–45% external rotation loads for jump-stop<br>• No specific MC phases predisposes women to greater risk, rather the phases produce variations in knee laxity, with increases in laxity increasing multiplanar knee joint loads and potential ACL injury risk.<br>• Higher knee laxity associated with increased multiplanar knee joint loads. However, variation in the MC phase where the highest and lowest laxity observed.<br>• Thus, group analysis is potentially masking differences. Clear individual variation in laxity and subsequent knee joint loading | Examined two tasks which is positive. Specific sport, skill, or strength training history not stated. Non-randomised testing order which introduces potential order effect. 3D motion analysis using limited marker set. Only one cutting angle examined at a low entry speed. Limited number of biomechanical variables in relation to ACL examined; only knee kinematics; ACL injury multi-segment mechanism (i.e., hip, trunk and foot). No temporal analysis / coordination. Group analysis masked out differences due to positive and negative responders. No reliability measures reported for outcome measures. Early follicular phase slightly spans later follicular MC phase classification according to McNulty et al. [24]. No sEMG. Changes were not examined relative to measurement error. |

*(Continued)*

**Table 4.** (Continued)

| Author | Sample size<br>n = Population details<br>(i.e., sport, playing level,<br>age, height)<br>Population training,<br>strength, skill history | Population menstrual / health<br>status and length<br>MC phase method verification<br>and details | Dynamic task and methods<br>Key outcome measures /<br>variables<br>Reliability / familiarisation<br>reported (y/n) | Results and implications (i.e.,<br>does a certain MC phase<br>predispose athlete to greater risk<br>of injury? | Comments |
|---|---|---|---|---|---|
| Park et al. (2009) BJSM | n = 25<br>Age: 22.7 ± 3.5 years, 170.2 ± 7.0 cm, 64.7 ± 9.6 kg<br>Sports at recreational level—activity levels 8.7 ± 4.6 hr/week<br>Specific sport, skill, or strength history not stated. | MC length: 28.9 ± 2.8 days<br>Follicular (5–8) (Spans both early and late according to McNulty's et al's [24] classification), ovulation (24–48 hours after detection of oestrogen surge—ovulation kit) and luteal (~7 days post 2nd test; (mid-luteal))—serum hormone. Non-randomised order. Blood samples collected at 3 time points to verify. Participants non-HC usages ≥ 3months. | 45° pre-planned cut; target speed 3.5 m/s. 3D motion and GRF analysis<br>Knee laxity—KT-2000 arthrometer 89N<br>Knee laxity (mm),peak knee angle and knee joint moment (Nm) and knee joint impulse (Nms). Internal knee adduction, knee flexion, and int/ext rotation moment during stance phase.<br>No familiarisation or reliability measures reported for outcome measures. | • ↑ knee laxity in ovulation vs. mid-luteal phase, but ↔ knee joint mechanics between MC phases.<br>• Positive correlation between Δ knee laxity and Δ knee joint loads (moment or impulse) from follicular phase to ovulation (r = 0.523), and ovulation to mid-luteal luteal phase (r = 0.526)<br>• Positive correlations between Δ laxity and Δ knee internal rotation impulse, and Δ laxity and Δ knee adduction impulse from ovulation to the luteal phase (r = 0.450) and (r = 0.408)<br>• 15 women with ↑ knee laxity from follicular to ovulation phases showed tendency to ↑ knee joint loads (adduction / impulse), whereas other 10 with ↓ knee laxity, showed a tendency to ↓ knee joint loads.<br>• 20 women whose knee laxity ↓ from ovulation to mid-luteal phase showed a tendency to ↓ knee joint loads (internal rotation moment and impulse and adduction impulse), whereas the other five, whose knee laxity ↑, showed a tendency to ↑ knee loads.<br>• Women whose knee laxity ↑ between MC phases showed ↑ knee joint loads, and women whose knee laxity ↓ showed ↓ knee joint loads.<br>• 1–3 mm Δ knee laxity during the MC caused a 3–4 Nm Δ internal rotation moments and 40–50 Nm Δ adduction moments.<br>• No specific MC phase predisposes females to greater risk of ACL injury based on biomechanical risk injury risk surrogates.<br>• Changes in laxity associated with change in KJL—due to MC phase hormonal changes (positive = positive / negative = negative). However, considerable variability in changes (magnitude and direction) in laxity across MC phases. | Specific sport, skill, or strength training history not stated. Non-randomised testing order. 3D motion analysis with limited marker set. Only one cutting angle examined at a low entry speed. Limited number of biomechanical variables in relation to ACL injury risk examined; only knee kinematics; ACL injury multi-segment mechanism (i.e., hip, trunk and foot). No temporal analysis / coordination. Group analysis masked out differences due to positive and negative responders. No reliability measures reported for outcome measures. Follicular phase spans early and later follicular MC phase classification according to McNulty et al. [24] No sEMG. Changes were not examined relative to measurement error |

*(Continued)*

**Table 4.** (Continued)

| Author | Sample size<br>n = Population details<br>(i.e., sport, playing level,<br>age, height)<br>Population training,<br>strength, skill history | Population menstrual / health<br>status and length<br>MC phase method verification<br>and details | Dynamic task and methods<br>Key outcome measures /<br>variables<br>Reliability / familiarisation<br>reported (y/n) | Results and implications (i.e.,<br>does a certain MC phase<br>predispose athlete to greater risk<br>of injury? | Comments |
|---|---|---|---|---|---|
| Shultz et al. (2012) | n = 71<br>Recreationally active women (2.5–10 hrs/week) for 3 months prior. Other anthropometrics not reported.<br>Specific sport, skill, or strength training history not stated. | Nulliparous—self reported MC length (26–32 days) ± 1 day<br>Ovulation kit to verify ovulation. Early follicular (up to 6 days after menses) and mid-luteal (8–10 consecutive days after ovulation).<br>Testing performed on days based on two time points where knee laxity was highest (T2) and lowest (T1) which coincided with these periods. Participants non-HC usages ≥ 3months. | Bilateral drop-jump from 0.45 m. 3D motional and GRF analysis with sEMG (quadriceps, hamstrings, gastrocnemius).<br>Knee laxity—KT-2000 arthrometer 133N. Electromagnetic position sensors (Ascension Technology Corp, Burlington, VT) 10 Nm varus/valgus torque and 5Nm internal-external torque)<br>Sagittal, frontal, and transverse kinetics and kinematics.<br>Muscle pre- landing and post-landing activation. Full temporal analysis. Participants' familiarisation 2 weeks prior to testing. Reliability for laxity measurements and landing duration only—male data only | • Sub-group of women who ↑ both sagittal and frontal plane laxity from T1 to T2 (cluster 3 and 4) had greater relative net change toward knee valgus of 3.7-to 5.2˚ compared with clusters who did not increase sagittal and frontal plane laxity (cluster 1).<br>• ↔ in muscle activation and moments between time points.<br>• Women who demonstrate changes in anterior and frontal laxity may display more valgus motion; however, moments and muscle activity do not change across MC phases which ultimately contributes to ACL loading. | Specific sport, skill, or strength training history not stated. Large sample size which is positive. No reliability measures reported for outcome measures. Temporal analysis conducted which is a positive of study. Unclear if testing order was randomised. MC phases not verified with blood samples. Changes were not examined relative to measurement error |

Key: Anterior cruciate ligament; GRF: Ground reaction force; sEMG: Surface electromyography; MC menstrual cycle; 3D: Three-dimensional; GM: Gluteus maximus; ST: Semitendinosus; BF: Biceps femoris; RF: Rectus femoris; SD: Standard deviation; NMS: neuromuscular; CUT45: 45˚ pre-planned cut. ↑: significantly greater / higher / increase ($p < 0.05$); ↓: significantly lower / lesser / decrease ($p < 0.05$); ↔ no significant change or difference ($p > 0.05$)

[35, 64] showed evidence that the mid-luteal phase may predispose women to greater risk of non-contact ACL injury compared to early or late follicular phases. Importantly, the MC influenced knee laxity [33, 34, 63], with two studies [33, 34] demonstrating MC attributed changes in knee laxity were associated with changes in KJL (i.e., increase in laxity associated with increase in KJL) and thus potential ACL injury risk [9, 47, 52]. However, considerable individual variation (i.e., magnitude and direction) was observed with respect to the MC phase which elicited the greatest knee laxity and KJL [33, 34]. Finally, the research in this review was low to very low in methodological quality, with significant methodological and research design limitations which should be acknowledged when interpreting the findings and improved in future research. A comprehensive overview regarding the methodological and research design limitations, considerations, and recommendations for future research are presented in Table 5.

## 4.1 Evidence showing MC phase has no effect on ACL neuromuscular or Neuromuscular injury risk surrogates

Four studies [33, 34, 62, 65] reported no significant differences in neuromuscular or biomechanical (anterior tibial shear, KJL or knee valgus) ACL injury risk surrogates between MC phases during jump-landing. Similarly, two studies revealed no biomechanical differences in knee injury risk surrogates during 45˚ cutting [33, 34] or stop-jump actions [33] between MC phases. Thus, in line with GRADE interpretation, there is very low level of evidence which potentially implies that no specific MC phase elevates non-contact ACL injury risk based on

**Table 5. Limitations and considerations of the synthesised research pertaining to the effect of the MC on ACL neuromuscular and biomechanical injury risk surrogates.**

| Limitations and considerations | Recommendations |
|---|---|
| Sample sizes are generally low and not justified (*n* = 10–28) [33–35, 62, 64, 65] with no *a priori* statistical power or sample size estimates calculated. Studies are likely underpowered which increases the risk of type II error [91], significant effects are likely to overestimate population effects sizes [91], have lower precision [91], and are less likely to be true and reproducible [92]. | Greater sample sizes with *a priori* sample size estimation and rationale are needed to ensure study is sufficiently powered. Researchers are advised to explicitly state the method for calculating statistical power (software, inputs etc.) and specify the chosen test to delineate the given effect size [91] or clearly state the sample size estimation method. |
| Participant characteristics are generally poorly described and limited to recreational or physically active women. Across all studies, none provided any information pertaining to specific sports, skill level, or resistance training history. This omission is key because skill level, training history, and physical capacity can influence movement strategy [93–97]. Additionally, not all women are the same, as more experienced naturally menstruating / eumenorrheic sportswomen can have different hormonal profiles concentrations compared to untrained women which is likely to influence physiology and subsequent performance [98]. | Better reporting of participant characteristics, particularly in relation to specific sports, skill level, or resistance training history [93], and further investigations into the effect of MC on ACL injury risk surrogates in a range of different sporting populations and skill levels is needed, particularly in elite populations. |
| Most studies investigated jump-landing tasks, using generally one or two tasks to evaluate potential injury risk. Additionally, only two studies examined a COD task, but this was limited to a 45˚ side-step cutting task only [33, 34]. Athletes' biomechanical injury risk profiles are task dependent [79–82] and importantly, a wide range of COD actions are performed in sport [99, 100] which are linked to ACL injuries including crossover cuts, split steps, shuffle steps, and pivots / turns. Additionally, the biomechanical demands of COD are angle and velocity dependent [80, 101], which have distinct implications on the technical execution, muscle activation, kinetic and kinematics, and KJLs. Thus, investigating only one COD angle only provides insight into an athlete's ability to COD at the specific angle and action. No study examined penultimate foot contact braking strategies which can play an important role in COD [102, 103], and no study examined a deceleration task: a key mechanism of ACL injury [104–107]. | Further research using a range of different tasks such as jump-landing, COD, and deceleration actions, with considerations of the angle-velocity trade-off during COD. |
| Majority of studies have focused on lower-limb biomechanics, particularly the knee, and have failed to explore the hip, trunk, and ankle kinetics or kinematics [33, 34, 62, 65]. This is important because the ACL injury mechanism and loading is multi-segmental [46, 52, 77, 78]. | Comprehensive whole-body analysis is needed when exploring the effect of the MC on neuromuscular and biomechanical ACL injury risk surrogates. |
| Three studies examined neuromuscular activity using sEMG techniques during abovementioned tasks [35, 63, 64]. Alternatively, high density sEMG can provide more detailed insights into the spatial distribution of activity and muscle coordination which can help understand how specific neuromuscular activity patterns affect movement quality, strategy, and potential KJLs. | Further research is needed using high density sEMG in addition to bipolar, which can provide novel insights into whether the MC alters the muscle coordination and spatial distribution of activity which can therefore affect movement quality, strategy, and potential knee joint loading and subsequent ACL injury risk. |
| Only one study randomised the order of testing across MC phases [64], only one study clearly stated that testing conditions (extraneous factors) were standardised between MC phases [65], while one study confirmed that the examiner was blinded to MC phase [35]. The lack of randomisation introduces the potential for order / learning / familiarisation effects, while the lack of blinding can introduce issues with bias. The lack of testing standardisation (extraneous factors) between MC phases can influence physiological functions and concentrations of reproductive hormones attributed to changes in circadian rhythm (time of day), diet and nutritional supplementation, prior exercise, alcohol and consumption and smoking [25, 29]. | Future research should ensure that: 1) randomised or counterbalanced testing order is adopted; 2) that the assessor is blinded to the MC phase; and 3) ensure that extraneous factors are standardised when conducting repeated measures / longitudinal testing. |
| Majority of studies fail to conduct individual analysis, and generally determine the effect of the MC on neuromuscular and biomechanical injury risk surrogates based on group analysis. Because not all women are the same, and the MC can result in inter-individual variations in physiological and hormonal responses [25–27], inferences based on group means only may conceal potentially meaningful information [108–110]; thus, the monitoring of individual changes and identification of positive-, non- and negative responders, would provide further insight into the effect of the MC on injury risk and performance. These individual changes should be interpreted relative to measurement error, SDD, or SWC as adopted in recently published literature [110–112]. Finally, most data are presented in tabular or column chart format, and individual data is not provided in figures. This reduces data transparency, and does not allow researchers to examine the variability and distribution within the data, identify outliers, and examine trends in the individual changes in outcome variables between the MC phases [113]. This is of particular importance when working with small sample sizes. | In addition to group analysis, researchers are encouraged to conduct individual analysis, monitoring positive, negative and non-responders. Ideally, when monitoring changes, these should also be interpreted relative to measurement error, SDD, or SWC. Finally, data transparency is advocated, and researchers should present their individual data, such as using univariate scatter plots. |

*(Continued)*

**Table 5.** (Continued)

| Limitations and considerations | Recommendations |
|---|---|
| No study reported reliability measures for their biomechanical or neuromuscular data, and only two studies stated a familiarisation session(s) / period [62, 63] was performed. Reliability is central in order to establish confidence in the data collection, analysis, and participant's performance of the task, and to determine whether the data is stable, consistent, accurate, and valid. Unfortunately, most investigations failed to report reliability measures, and evidence has shown that injury risk surrogates, such as KJLs, can be susceptible to variability between sessions [114–116]. | Ensure participants are adequately familiarised with the testing protocols and ensure the reliability measures are reported to ensure data collected is stable, accurate, and valid. This will also result in greater data transparency and research quality. |
| No study examined joint-joint coordination changes between MC phases [33–35, 62–65]. Examinations of joint-joint coordination; angle-angle plots; position-velocity plots, will provide further insight into coupling behaviours and movement strategy and execution during high intensity tasks [117–119]. Only one study included a form of temporal analysis [63] while the remaining studies generally conducted discrete point analysis [33–35, 62, 64, 65]. This approach can lead to regional focus bias, whereby a large amount of data are discarded from the entire waveform [120–124]; thus, valuable information across the whole curve is unexamined because only a single data point is examined [125, 126]. Additionally, discrete point analysis does not consider the position of the key measures (i.e., differences in timing); for example, trial peaks may occur at different timings along the waveform and thus, the temporal organisation of the pattern is lost [127–129]. | Future research should consider examining coordination changes in movement between MC phases to establish if the MC alters movement strategy and coupling behaviours between multiple segments. In addition to discrete point analysis, future research should conduct full temporal analysis of the full waveform, using statistical approaches such as statistical parametric mapping [114, 130–132] or temporal phase analysis [133–135]. Conducting such analysis could provide further insight not only into the magnitude of differences, but importantly provide more detail into where these differences occur across the whole time-series. |

**Key:** COD: Change of direction; MC: menstrual cycle; ACL: anterior cruciate ligament; KJL: knee joint load; SWC: smallest worthwhile change; SDD: smallest detectable difference. sEMG: surface electromyography.

biomechanical surrogates. The lack of differences in biomechanical ACL injury risk surrogates between ovulation and mid-luteal phases, speculatively, could be attributed to similar, high concentrations of oestrogen which may have comparable effects on ligament and tendon properties, neuromuscular control, and thus potential injury risk [67].

Although Chaudhari et al. [65] standardised testing conditions when comparing biomechanics between MC phases, the phase descriptions were unclear (i.e., failed to specify day that MC phase coincided with); therefore, making it difficult to verify and ascertain whether accurate MC phases were identified [24, 25]. Additionally, Park et al. [33, 34] identified the early follicular phase, using a date range that spans both early and late follicular phase according to recent descriptions [24, 25]. This approach may lead to the grouping of non-homogeneous participants and potential inaccurate evaluation regarding the influence of the MC [25]. The sample sizes used in the studies are generally small ($n$ = 10–28) and likely underpowered. Two studies stated the testing order was non-randomised [62, 65], while two studies did not clearly describe whether randomised testing occurred [33, 34]. Thus, the lack of differences observed between MC phases by researchers [33, 34] could be influenced by a learning or order effect, potentially confounding the observations. Based on current evidence, methodological, and research design limitations, practitioners should be cautious manipulating their injury mitigation, screening, and physical preparation strategies based on the MC for female athletes.

### 4.2 Evidence showing mid-luteal phase may increase ACL injury risk based on neuromuscular or biomechanical injury risk surrogates

Two studies [35, 64] indicated that the mid-luteal phase may predispose women to greater non-contact ACL injury risk compared to other phases based on biomechanical (i.e., kinematics linked to greater KJLs) [35] and neuromuscular (i.e., reduced gluteal activation [35] or delayed semitendinosus activation which could increase anterior tibial shear [64]) surrogates

during drop-landing tasks. This delay, speculatively, could be attributed to significant increases in progesterone compared to other phases (i.e., follicular and ovulation) which can elicit nervous system inhibitory effects [68, 69]. Importantly, however, minimal differences were observed in onset timing for other lower-limb muscles by Dedrick et al. [64]. Nonetheless, these studies have either blinded the assessor [35] or randomised testing [64], thus reducing bias.

Okazaki et al. [35] and Dedrick et al. [64] did not examine KJLs, a key ACL loading surrogate [48, 70–72]. Akin to previous studies [33, 34, 62, 65], reliability measures were not reported for the outcome measures, while changes in measures were not interpreted relative to measurement error. This raises questions as to whether observed muscle activity [64] and joint kinematic [35] changes were due to error (i.e., neuromuscular or movement variability and measurement error) or truly attributable to MC influenced hormonal changes. Further research is needed which examines the effect of the MC on neuromuscular and biomechanical ACL injury risk surrogates, particularly KJLs, accounting for measurement error when interpreting changes between MC phases.

## 4.3 Evidence supporting an effect of MC on knee laxity and subsequent knee joint loads

While the MC's effect on neuromuscular and biomechanical ACL injury risk surrogates appears inconclusive, three studies [33, 34, 63] demonstrated changes in knee laxity between different MC phases which were accompanied with changes in potentially hazardous biomechanics associated with ACL loading. Park et al. [33, 34] reported no differences in KJLs between MC phases, but measured knee laxity at each MC phase. Interestingly, women with higher knee laxity [33], and increases in laxity between MC phases [34], were associated with greater KJLs during cutting or stop-jumping. These findings [33, 34] may help explain why prospective research has identified an association between knee laxity and non-contact ACL injury [73, 74]. However, the MC phase which produced the greatest knee laxity and subsequent KJL was inconsistent between women (see Table 4), and thus an individualised approach to laxity, neuromuscular, and biomechanical monitoring is advised.

Collectively, this research indicates no specific MC phase predisposes women to greater risk of ACL injury based on biomechanical injury risk surrogates or knee laxity [33, 34]; however, the changes in laxity associated with MC phase hormonal changes produces a problematic laxity effect for KJLs and potential ACL injury risk. Notably, considerable variability in the changes (i.e., magnitude and direction) in laxity and its effect on KJLs was observed across MC phases which could be attributed to intra- and inter-individual-variation in the magnitude and rate of change in ovarian hormonal concentrations which influences laxity [75]. Additionally, genetic variation between individuals and differential expression of oestrogen receptors may effect oestradiol's ($E_2$) ability to bind to its receptor, potentially varying oestrogen-attributed physiological responses in musculoskeletal connective tissue, thus laxity responses [76].

Caution is advised regarding the aforementioned research [33, 34] as some definitions and classifications for the early follicular phase are different to recent MC classifications [24, 25], spanning both early and late phases (i.e., 5–8 days and 3–7 days). Additionally, similar to aforementioned studies [33, 34, 62, 65], only knee joint kinetics and kinematics were examined [33, 34], with other segments and joints (trunk, hip, and ankle), and neuromuscular activity also unexamined. This absence is important because the ACL injury mechanism and loading is multi-segmental [46, 52, 77, 78], and neuromuscular activity patterns has the potential to unload [55] or increase ACL loading [56–59]. Conversely, in a larger sample size, previous work [63] reported that women who demonstrate changes in anterior and frontal knee laxity

may display more valgus motion during bilateral drop vertical jumping; however, KJL and muscle activity did not change across phases which ultimately contributes to ACL loading and injury risk [47, 55]. Although difficult to rationalise these contrasting observations with Park et al. [33, 34], athletes' biomechanical injury risk profiles are task dependent [79–82]; thus, caution is advised generalising the conclusions regarding the role of the MC and biomechanical and neuromuscular injury risk surrogates when only a limited number of tasks have been investigated.

Although there is no consistent MC effect on knee laxity, MC hormonal perturbations can both increase and decrease women's knee laxity at each MC phase. Therefore, monitoring female athletes' knee laxity changes could be a viable strategy to infer potential KJLs changes and potential ACL injury risk throughout the MC. This can be done using an arthrometer [33, 34, 83, 84], rolimeter [85, 86], ultrasound [87], radiography [88], or wearable accelerometers [84].

## 5. Limitations, considerations, and future recommendations for research

Recently suggested [75], greater emphasis should be placed on exploring the effect of hormonal concentrations (i.e., magnitude, relative and / or rate of change from baseline) rather than focusing on the MC phase's effect on performance or injury risk, because the change in hormonal contribution ultimately affects the physiological system. Further research is needed to understand how hormonal, neuromuscular, and biomechanical ACL injury risk factors interrelate and influence joint laxity and movement execution of dynamic tasks at different MC phases, whilst considering hormonal concentrations.

There is significant underrepresentation of female athletes in sports and exercise medicine research [23, 89, 90]. Researchers often avoid investigating women due to the MC associated physiological changes and the potential methodological difficulties [25]. The limited published literature synthesised during this review is insightful and provides unique, important information from an underrepresented population. We have, however, highlighted some methodological and research design limitations, and have suggested some future recommendations for research to build on the insightful body of work to improve research quality. Consequently, this has led to recent recommendations for more research into female athletes [23, 25, 26], particularly more rigorous research designs when exploring the MC's effect in relation to potential injury risk and exercise performance [25, 26]. Therefore, future investigations which follow this review and other researchers' suggestions [25, 26, 29], accounting for the methodological and research design limitations, will produce greater methodological quality and higher-quality data in women. This will permit fairer and more accurate conclusions regarding the MC's effect on ACL injury risk.

Strengths of this systematic review included the comprehensive search strategy conforming with PRISMA, adopting the PICOS strategy to permit the synthesis of methodology and study findings. Additionally, methodological quality was assessed using a modified Downs and Black checklist [24], though this version has not been validated. Overall quality of evidence was evaluated using the GRADE approach, but due to the heterogeneity, quantitative statistical analysis and a meta-analysis could not be performed. Finally, this review was not pre-registered which can increase risk of bias (i.e., collating and synthesis of research, selective reporting, overall transparency, duplication, research waste).

## 6. Conclusion

Based on this review, it is inconclusive whether a particular MC phase predisposes eumenorrheic and naturally menstruating women to greater non-contact ACL injury risk based on

neuromuscular and biomechanical surrogates, with mixed findings observed. Interestingly, knee laxity was affected by the MC, with evidence that MC attributed changes in knee laxity were associated with changes in KJL and thus potential ACL injury risk. However, considerable individual variation (i.e., magnitude and direction) was observed with respect to the MC phase which elicited the greatest knee laxity and KJL. Nonetheless, monitoring changes in knee laxity in female athletes could be a viable strategy to infer potential changes in KJLs and ACL injury risk. Research synthesised in this review was low to very low in methodological quality, contributing to a very low quality of evidence, which could be improved with respect to design and execution. As such, it is difficult to make definitive conclusions regarding the effects of the MC phase on ACL neuromuscular and biomechanical injury risk surrogates, and thus practitioners should be cautious manipulating their physical preparation, injury mitigation, and screening practises based on current evidence.

## Supporting information

**S1 Checklist. PRISMA 2020 checklist.**
(TIF)

## Author Contributions

**Conceptualization:** Thomas Dos'Santos, Georgina K. Stebbings, Andy Sanderson.

**Formal analysis:** Thomas Dos'Santos, Medha Shashidharan, Katherine A. J. Daniels, Andy Sanderson.

**Investigation:** Thomas Dos'Santos, Medha Shashidharan, Andy Sanderson.

**Methodology:** Thomas Dos'Santos, Georgina K. Stebbings, Christopher Morse, Medha Shashidharan, Andy Sanderson.

**Project administration:** Thomas Dos'Santos.

**Writing – original draft:** Thomas Dos'Santos, Georgina K. Stebbings, Christopher Morse, Katherine A. J. Daniels, Andy Sanderson.

**Writing – review & editing:** Thomas Dos'Santos, Georgina K. Stebbings, Christopher Morse, Medha Shashidharan, Katherine A. J. Daniels, Andy Sanderson.

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
