## [Decision Letter · Decision Letter 0]

1 Sep 2022

PONE-D-22-07730

Effects of The Menstrual Cycle Phase on Anterior Cruciate Ligament Neuromuscular and Biomechanical Injury Risk Surrogates in Eumenorrheic and Naturally Menstruating Women: A Systematic Review and Recommendations for Future Research

PLOS ONE

Dear Dr. Dos'Santos,

Thank you for submitting your manuscript to PLOS ONE. After careful consideration, we feel that it has merit but does not fully meet PLOS ONE’s publication criteria as it currently stands. Therefore, we invite you to submit a revised version of the manuscript that addresses the points raised during the review process.

We look forward to receiving your revised manuscript.

Kind regards,

Kristian Samuelsson, MD, PhD, MSc

Academic Editor

PLOS ONE

https://journals.plos.org/plosone/s/file?id=ba62/PLOSOne_formatting_sample_title_authors_affiliations.pdf".

Reviewers' comments:

Reviewer's Responses to Questions

**Comments to the Author**

1. Is the manuscript technically sound, and do the data support the conclusions?

Reviewer #1: Yes

Reviewer #2: Yes

2. Has the statistical analysis been performed appropriately and rigorously? 

Reviewer #1: N/A

Reviewer #2: N/A

3. Have the authors made all data underlying the findings in their manuscript fully available?

Reviewer #1: Yes

Reviewer #2: Yes

4. Is the manuscript presented in an intelligible fashion and written in standard English?

Reviewer #1: Yes

Reviewer #2: Yes

5. Review Comments to the Author

Reviewer #1: In the manuscript ‘’Effects of The Menstrual Cycle Phase on Anterior Cruciate Ligament Neuromuscular and Biomechanical Injury Risk Surrogates in Eumenorrheic and Naturally Menstruating Women: A Systematic Review and Recommendations for Future Research’’, the authors attempted to address whether the menstrual cycle has an impact on the neuromuscular and biomechanical risk surrogates of ACL injury patients. Furthermore, the authors also provided comments on the current literature and propose suggestions for future research on the menstrual cycle and ACL injuries.

Although the manuscript may be of interest for the readers of PLOS ONE, it suffers some important issues.

General comments:

The manuscript is written with a high standard of English language. This promotes good readability and a general flow. Moreover, the article adheres to appropriate reporting guidelines for the current systematic review as it was conducted according to the PRISMA protocol. However, the manuscript in its current form is too long and there are some concerns with regards to reporting of the results.

Specific comments:

Title page

1. The title of the article is somewhat long, perhaps it would be useful to shorten it, for example: ‘’Effects of the menstrual cycle phase on anterior cruciate ligament injury risk surrogates in menstruating women: a systematic review’’.

2. Line 20: please change ‘’1 figures’’ to ‘’1 figure’’.

Abstract

1. Lines 43-48, it would be helpful to add which quality assessment tool(s) was used for the included studies of the systematic review. This could also be added to the last sentence of the results section (line 53) where the overall study quality of included studies is presented.

2. In the background, only the primary aim of the study is provided while in the conclusion, both the primary and secondary aims are listed. The background and the conclusion section should match in the abstract.

Introduction

1. The first, second, third and fourth paragraph of the introduction are some general aspects of ACL injuries which are already well known. These paragraphs may perhaps be removed from the manuscript so that the focus lies on the menstrual cycle and its relationship to ACL injuries.

2. Lines 171-75, the primary objective of the manuscript is provided twice although in different forms. I would suggest that the question ‘’does the way women perform the aforementioned high-risk manoeuvres predispose them to potentially greater non-contact ACL injury risk based on neuromuscular and biomechanical surrogates of non-contact ACL injury risk at a particular MC stage?’’ is removed from this paragraph.

Methods

1. Line 179 – the study protocol was not pre-registered, however – were the review methods established prior to the conduct of the review? Clarifying this to the reader would be helpful.

2. Table 1. The authors use references in the PICOS section, which in my opinion is not necessary. In the last row, the authors provide the data that was extracted. Was the data extraction process performed in duplicate?

3. Line 230 – it is stated that all studies in this review were observational although in Table 1 ‘’study design’’ only experimental studies were included. I believe that these 2 study designs are not equivalent to each other as observational studies can for instance be cohort studies while experimental studies can be performed in laboratories.

Results

1. Line 243 – it appears as if the table numbers have been written in the wrong order as table 2 includes the quality assessment according to Downs and Black checklist and not the study characteristics. Please revise the order of the tables. Additionally, it is stated that the sample sizes ranged from 10-28 although one study (Shultz et al.) had a sample size of 71 which leads to a range of 10-71.

2. Line 290-297 – the a-priori quality should be provided first, followed by the revised quality scores/quality evidence, i.e. the Downs and Black checklist scores, followed by the GRADE.

Table 3

1. The table includes valuable information of the study designs but is somewhat big, perhaps divide this table into 2 separate ones?

Discussion

1. The discussion in its current form is too long. For example, each paragraph includes detailed information on specific studies, perhaps these could be shortened?

2. The discussion section refers to the included tables of the manuscript which should be removed, for instance see line 335.

3. A comment on the revised Downs and Black checklist used in the current manuscript would be welcomed, is it validated for instance?

4. The discussion section is currently lacking a strength/limitation section of the methodology applied in the systematic review.

5. A short paragraph addressing the biochemical analyses and ovulation kits used to determine the phase of the menstrual cycle of included studies would be valuable to the manuscript.

Author contributions

The author contributions are stated in the methods section as all in the author contribution section, perhaps these could be listed in only one paragraph?

Reviewer #2: Dear authors,

As a reviewer, I appreciate your scientific efforts in elaborating the topic of the submitted study. The topic of the study is interesting und not well studied so far. The reviewer believes that the submitted study is of interest to the readership of PLOS ONE. However, there are several limitations that should be addressed before the study can be considered for publication in PLOS ONE.

The main limitations / revisions needed are as follows:

-The clinical relevance of the study needs to be highlighted. Why is this study needed and how is this study useful in daily clinical practice / research.

- Data analysis is poorly described and quantitative data were rarely reported. This can be done better.

-The manuscript is well written. However, the manuscript is far too long and needs to be shortened.

Title

The reviewer recommends to delete “and recommendations for future research”.

Abstract

Background: Adequate.

Methods: Line 48: The reviewer recommends to add one sentence about data analysis (e.g., quantitative, qualitative etc.).

Results: Line 53: Please be more specific. Provide quantitative data for “low to very low” study quality (i.e., modified Downs and Black checklist).

Conclusions: The reviewer recommends to shorten the Conclusion section of the abstract. In addition, add one sentence about the clinical relevance of this study.

Level of evidence: Is missing, please add.

Key Words: Adequate.

Introduction

Far too long. Please shorten and focus on the topic of the study. Why is this study needed? Highlight particular controversies in the introduction section. Adequate objectives. Hypothesis of the study is missing. Please add.

Line 65, 121: Define abbreviations when they are used for the first time (i.e., ACL, MC).

Material and Methods

There are several missing items in the material and methods section that should be added:

-Who performed data extraction?

-Which data were collected from each study for data synthesis?

-Statistical analysis and methods of data synthesis

Line 179-180: Why was this review not registered in advance? This is a major limitation and should be mentioned in the limitations section. This reduces the quality of the systematic review.

Line 187: It is recommended to add the full search strategy as a supplemental file.

Line 219-225: This is somewhat confusing. Based on Table 1, only “studies that verified participants’ MC phase via biochemical analysis (i.e., blood/serum analysis) and / or ovulation kits were included, in line with McNulty et al.”. Therefore, no study should have been downgraded? Please clarify.

Results

In general, it would be good to present a quantitative data synthesis. However, the studies may have been so heterogeneous that data synthesis was not possible, which is a limitation of the study. Please comment.

Discussion

The discussion section is well written. However, the discussion section is far too long. Please try to shorten. Focus on the topic of the study and highlight the clinical relevance of this study. How can the results of this study be useful in daily clinical practice?

Line 329: It is recommended to start the discussion section as follows: “The most important finding of this study was …”

Conclusion

Far too long. Please shorten. In the conclusion section, the authors should “conclude” their findings in 2-5 sentences. In addition, the clinical relevance of the study should be highlighted. Moreover, it is recommended to delete quotations (both for references and tables) in the conclusion section.

Figures

Figure 1: Adequate.

Tables

Table 1: How did the authors confirm that the women were eumenorrheic? Please clarify.

Table 1: Please be more specific and define biomechanical and/or neuromuscular surrogates.

Table 2: Adequate.

Table 3: Adequate.

Table 4 + 5: Exhaustive and busy table. Try to shorten.

References

Adequate.

6. PLOS authors have the option to publish the peer review history of their article (what does this mean?). If published, this will include your full peer review and any attached files.

Reviewer #1: No

Reviewer #2: No

---

## [Author Response · Author response to Decision Letter 0]

21 Sep 2022

General comments:

The manuscript is written with a high standard of English language. This promotes good readability and a general flow. Moreover, the article adheres to appropriate reporting guidelines for the current systematic review as it was conducted according to the PRISMA protocol. However, the manuscript in its current form is too long and there are some concerns with regards to reporting of the results.

Response: Thank you for your comment. We have done our best to reduce the word count, reducing the word count overall from 6777 words to 5728. This has included some extra additions in the methods and discussion as your requests and reviewer 2. 

Specific comments:

Title page

1. The title of the article is somewhat long, perhaps it would be useful to shorten it, for example: ‘’Effects of the menstrual cycle phase on anterior cruciate ligament injury risk surrogates in menstruating women: a systematic review’’.

Response: Thank you for your comment. We agree that it is slightly long; however, we believe it is more appropriate and important to use the scientific terminology as recommended by Elliott-Sale et al. (2021) to specifically provide context regarding the reproductive profiles of the women of this systematic review (i.e., Eumenorrheic and Naturally Menstruating). Moreover, there are a plethora of different ACL risk factors and surrogates; however, we are reluctant to remove the terms “neuromuscular and biomechanical” because we feel the inclusion of there terms provides more context and scope for the systematic review. However, in light of your comment and R2’s, we have removed the term “and recommendations for future research” from the title.

2. Line 20: please change ‘’1 figures’’ to ‘’1 figure’’.

Response: Thank you for your comment. This has been amended.

Abstract

1. Lines 43-48, it would be helpful to add which quality assessment tool(s) was used for the included studies of the systematic review. This could also be added to the last sentence of the results section (line 53) where the overall study quality of included studies is presented.

Response: Thank you for your comment. This has been amended. Please see line 50-53 and 58.

2. In the background, only the primary aim of the study is provided while in the conclusion, both the primary and secondary aims are listed. The background and the conclusion section should match in the abstract.

Response: Thank you for your comment. This has been amended where we have added a brief indication of the secondary aim. Please see line 42.

Introduction

1. The first, second, third and fourth paragraph of the introduction are some general aspects of ACL injuries which are already well known. These paragraphs may perhaps be removed from the manuscript so that the focus lies on the menstrual cycle and its relationship to ACL injuries.

Response: Thank you for your comment. We have reduced the introduction by 2 paragraphs / 451 words (1780 to 1329 words) and we hope that the introduction is more concise. We decided that the content related to the biomechanical and neuromuscular risk factors should stay, given the importance of these risk factors on potential mechanical loads and the fact that these risk factors were the inclusion criteria for the systematic review, thus providing context for the reader.

2. Lines 171-75, the primary objective of the manuscript is provided twice although in different forms. I would suggest that the question ‘’does the way women perform the aforementioned high-risk manoeuvres predispose them to potentially greater non-contact ACL injury risk based on neuromuscular and biomechanical surrogates of non-contact ACL injury risk at a particular MC stage?’’ is removed from this paragraph.

Response: Thank you for your comment. This has been amended and we have removed your suggested sentence. Please see line xx.

Methods

1. Line 179 – the study protocol was not pre-registered, however – were the review methods established prior to the conduct of the review? Clarifying this to the reader would be helpful.

Response: Thank you for your comment. This has been amended and we have added the following: however, the review methods were established prior to conducting the review. Please see line 153.

2. Table 1. The authors use references in the PICOS section, which in my opinion is not necessary. In the last row, the authors provide the data that was extracted. Was the data extraction process performed in duplicate?

Response: Thank you for your comment. With respect, would still like to include the references in table 1. We feel the inclusion of references further supports and justifies the PICOS strategy for specific population, method, comparator, or outcome criteria. Yes, performed in duplicate. This is added on Line 182. 

3. Line 230 – it is stated that all studies in this review were observational although in Table 1 ‘’study design’’ only experimental studies were included. I believe that these 2 study designs are not equivalent to each other as observational studies can for instance be cohort studies while experimental studies can be performed in laboratories.

Response: Thank you for your comment. Apologies for this error. This has been amended.

Results

1. Line 243 – it appears as if the table numbers have been written in the wrong order as table 2 includes the quality assessment according to Downs and Black checklist and not the study characteristics. Please revise the order of the tables. Additionally, it is stated that the sample sizes ranged from 10-28 although one study (Shultz et al.) had a sample size of 71 which leads to a range of 10-71.

Response: Thank you for your comment. Apologies, this has been amended. Please see line 225-228.

2. Line 290-297 – the a-priori quality should be provided first, followed by the revised quality scores/quality evidence, i.e. the Downs and Black checklist scores, followed by the GRADE.

Response: Thank you for your comment. This has been amended in light of your suggestions. Please see line 274-280.

Table 3

1. The table includes valuable information of the study designs but is somewhat big, perhaps divide this table into 2 separate ones?

Response: Thank you for your comment. While it is a quite a large table, we feel it is best to contain all of this information in one table so all information is collated in one area and table. If accepted, the journal publishers can also format the table to aid with clarity. 

Discussion

1. The discussion in its current form is too long. For example, each paragraph includes detailed information on specific studies, perhaps these could be shortened?

Response: Thank you for your comment. We have done our best to reduce the word count of the discussion. As such, we have reduced the length by 742 words, from 2882 to 2140 words.

2. The discussion section refers to the included tables of the manuscript which should be removed, for instance see line 335.

Response: Thank you for your comment. References to tables have been removed in the discussion.

3. A comment on the revised Downs and Black checklist used in the current manuscript would be welcomed, is it validated for instance?

Response: Thank you for your comment. We have commented on this on line 455.

4. The discussion section is currently lacking a strength/limitation section of the methodology applied in the systematic review.

Response: Thank you for your comment. We have added a small paragraph regarding the strength and limitations. Please see line 454-459. 

5. A short paragraph addressing the biochemical analyses and ovulation kits used to determine the phase of the menstrual cycle of included studies would be valuable to the manuscript.

Response: Thank you for your comment. We are conscious of the word count and are, with respect, reluctant to include further information. We do briefly discuss which studies used biochemical analysis / ovulation kits in the results section (line 249-251) and information is also presented in Table 4.

Author contributions

The author contributions are stated in the methods section as all in the author contribution section, perhaps these could be listed in only one paragraph?

Response: Thank you for your comment. Unfortunately, we do not fully understand this comment. We have followed the author contributions guidance as per PlosOne journal specifications. 

Reviewer #2: Dear authors,

As a reviewer, I appreciate your scientific efforts in elaborating the topic of the submitted study. The topic of the study is interesting und not well studied so far. The reviewer believes that the submitted study is of interest to the readership of PLOS ONE. However, there are several limitations that should be addressed before the study can be considered for publication in PLOS ONE.

The main limitations / revisions needed are as follows:

-The clinical relevance of the study needs to be highlighted. Why is this study needed and how is this study useful in daily clinical practice / research.

- Data analysis is poorly described and quantitative data were rarely reported. This can be done better.

-The manuscript is well written. However, the manuscript is far too long and needs to be shortened.

Response: Thank you for your comments. In the introduction, we high highlighted the clinical relevance of this study. Please see lines 142-145 and 151-153. We have now added a data analysis collations section. Please see line 212-217. We have done our best to reduce the word count, reducing the word count overall from 6777 words to 5728. This has included some extra additions in the methods and discussion as your requests and reviewer 1. We have reduced the introduction by 2 paragraphs / 451 words (1780 to 1329 words) and we hope that the introduction is more concise. Additionally, we have reduced the length of the discussion by 742 words, from 2882 to 2140 words.

Title

The reviewer recommends to delete “and recommendations for future research”.

Response: Thank you for your comment. This has been amended.

Abstract

Background: Adequate.

Methods: Line 48: The reviewer recommends to add one sentence about data analysis (e.g., quantitative, qualitative etc.).

Response: Thank you for your comment. This has been amended. We have added the following on line 49: “Quantitative data pertaining to study methodology, participant, MC phase and verification, ACL injury risk surrogates, reliability measures, and results were obtained for qualitative analysis.”

Results: Line 53: Please be more specific. Provide quantitative data for “low to very low” study quality (i.e., modified Downs and Black checklist).

Response: Thank you for your comment. This has been amended. Please see line 58 where we state the following: “Study quality and quality of evidence was low to very low (7-9 Modified Downs and Black Checklist score 7-9; and GRADE).”

Conclusions: The reviewer recommends to shorten the Conclusion section of the abstract. In addition, add one sentence about the clinical relevance of this study.

Response: Thank you for your comment. Due to the extensive nature of this systematic review, and complexities and caveats with the evaluation of the literature, we believe is an inappropriate to try and summarise this in one sentence. Previous systematic reviews have used multiple sentences in the abstract conclusion when contextualising their findings. Please see McNulty et al. 2021 and Somerson et al. 2019 in our reference list. We have amended some sentences to further highlight the clinical relevance of our review findings. Please see line 60-66.

Level of evidence: Is missing, please add.

Response: Thank you for your comment. This has been added. Please see line 67.

Key Words: Adequate.

Introduction

Far too long. Please shorten and focus on the topic of the study. Why is this study needed? Highlight particular controversies in the introduction section. Adequate objectives. Hypothesis of the study is missing. Please add.

Response: Thank you for your comments. In the introduction, we high highlighted the clinical relevance of this study. Please see lines 142-145 and 151-153. We have reduced the introduction by 2 paragraphs / 451 words (1780 to 1329 words) and we hope that the introduction is more concise. 

Hypotheses have also been added. Please see line 151: “It was hypothesised that differences in neuromuscular and biomechanical injury risk surrogates would be observed between MC phases in eumenorrheic and naturally menstruating women.”

Line 65, 121: Define abbreviations when they are used for the first time (i.e., ACL, MC).

Response: Thank you for your comment. This has been added. Please see line 69 and 96.

Material and Methods

There are several missing items in the material and methods section that should be added:

-Who performed data extraction?

-Which data were collected from each study for data synthesis?

-Statistical analysis and methods of data synthesis

Response: Thank you for your comment. We have now added a data collation section (2.4) which provides information pertaining to data extraction and specific data. No statistical analysis / meta-analysis was performed due to the heterogeneity in methodology and outcome measures.

Line 179-180: Why was this review not registered in advance? This is a major limitation and should be mentioned in the limitations section. This reduces the quality of the systematic review.

Response: Thank you for your comment. In hindsight, we should have registered the review. We have acknowledged this as a limitation. Please see line 460.

Line 187: It is recommended to add the full search strategy as a supplemental file.

Response: Thank you for your comment. The full search strategy is provided on line 173.

Line 219-225: This is somewhat confusing. Based on Table 1, only “studies that verified participants’ MC phase via biochemical analysis (i.e., blood/serum analysis) and / or ovulation kits were included, in line with McNulty et al.”. Therefore, no study should have been downgraded? Please clarify.

Response: Thank you for your comment. Yes, if they failed to do blood samples but used an ovulation kit, they were downgraded. Likewise, if they used a blood sample but no ovulation kits, then again, they were downgraded. Thus, to maintain their rating, they had to do both. This was the case for study 1, 2, 3, and 7 which were downgraded.

Results

In general, it would be good to present a quantitative data synthesis. However, the studies may have been so heterogeneous that data synthesis was not possible, which is a limitation of the study. Please comment.

Response: Thank you for your comment. Yes, we have now included a data collation section (2.4) and have acknowledged that that the data and methodologies were heterogeneous which precluded a meta-analysis. Please see line 281 and 460.

Discussion

The discussion section is well written. However, the discussion section is far too long. Please try to shorten. Focus on the topic of the study and highlight the clinical relevance of this study. How can the results of this study be useful in daily clinical practice?

Response: Thank you for your comment. We have reduced the length of the discussion by 742 words, from 2882 to 2140 words. We have further highlighted the clinical relevance in certain sections. Please see lines 345-347, 398, 439-442. However, because of the low quality of evidence and methodological quality, this review is primarily highlighting the future directions of research and methodological approaches to improve research quality and data to increase our understanding of the menstrual cycle on ACL injury risk. 

Line 329: It is recommended to start the discussion section as follows: “The most important finding of this study was …”

Response: Thank you for your comment. This has been amended. Please see line 305.

Conclusion

Far too long. Please shorten. In the conclusion section, the authors should “conclude” their findings in 2-5 sentences. In addition, the clinical relevance of the study should be highlighted. Moreover, it is recommended to delete quotations (both for references and tables) in the conclusion section.

Response: Thank you for your comment. We have reduced the length from 489 to 277 words and have removed quotations and references. 

Figures

Figure 1: Adequate.

Tables

Table 1: How did the authors confirm that the women were eumenorrheic? Please clarify.

This was based on the methodologies reported by the research group and interpreted according to the definition presented by Elliott-Sale et al. 2021 for eumenorrheic classification.

Table 1: Please be more specific and define biomechanical and/or neuromuscular surrogates.

Response: Thank you for your comment. This has been added. 

Table 2: Adequate.

Table 3: Adequate.

Table 4 + 5: Exhaustive and busy table. Try to shorten.

Response: Thank you for your comment. We feel these are critical tables for the systematic review and contain relevant and highly important information. In particular, Table 5 is central for highlighting the limitations and recommended areas for future research to improve research quality in this topic area. As such, with respect, we would like to keep the tables as they are. 

References

Adequate.

---

## [Decision Letter · Decision Letter 1]

24 Oct 2022

PONE-D-22-07730R1Effects of The Menstrual Cycle Phase on Anterior Cruciate Ligament Neuromuscular and Biomechanical Injury Risk Surrogates in Eumenorrheic and Naturally Menstruating Women: A Systematic ReviewPLOS ONE

Dear Dr. Dos'Santos,Thank you for submitting your manuscript to PLOS ONE. After receiving a new report from two external reviewers, both consider that your manuscript needs further revision. 

However, the opinion about the current version of the text is not uniform in both cases. Thus, after carefully reading the comments, I have recommended a major revision of the text. Please review all comments in detail and solve all the doubts raised about the content of your article

We look forward to receiving your revised manuscript.

Kind regards,

Javier Peña, Ph.D.

Academic Editor

PLOS ONE

Reviewers' comments:

Reviewer's Responses to Questions

**Comments to the Author**

1. If the authors have adequately addressed your comments raised in a previous round of review and you feel that this manuscript is now acceptable for publication, you may indicate that here to bypass the “Comments to the Author” section, enter your conflict of interest statement in the “Confidential to Editor” section, and submit your "Accept" recommendation.

Reviewer #1: All comments have been addressed

Reviewer #2: All comments have been addressed

2. Is the manuscript technically sound, and do the data support the conclusions?

Reviewer #1: Yes

Reviewer #2: Yes

3. Has the statistical analysis been performed appropriately and rigorously? 

Reviewer #1: N/A

Reviewer #2: N/A

4. Have the authors made all data underlying the findings in their manuscript fully available?

Reviewer #1: Yes

Reviewer #2: Yes

5. Is the manuscript presented in an intelligible fashion and written in standard English?

Reviewer #1: Yes

Reviewer #2: Yes

6. Review Comments to the Author

Reviewer #1: Abstract:

On line 59, please change (7-9 Modified Downs and Black Checklist score 7-9; and GRADE) so that it is understandable which score corresponds to which quality assessment/grading tool.

Line 65-66 – remove this sentence as it does not add additional value to the conclusion section.

Methods:

In table I ‘’study design’’, please change experimental to clinical or human.

Who performed the literature search in PubMed, Medline (OVID), SPORTDiscus, and Web of Science databases?

Line 169, please add the reference to the guideline that was used.

Results

I would recommend that ‘’statistically’’ is removed from ‘’no statistically significant’’ on line 255. Furthermore, what does ‘’meaningful difference’’ refer to on line 255?

Discussion

The sentence on line 305-308 is very long, I would suggest that the sentence is divided into two.

Please remove reference to table on line 454.

Line 461 – please add which type of bias(es) the review was subjected to.

Conclusion

The conclusion section is far too long. It needs to be shortened.

Reviewer #2: Dear authors,

The first revision substantially improved the quality of the study. However, the biggest limitation of the manuscript is that it is still far too long. In particular, the introduction, discussion, and conclusion sections need to be shortened. The authors should focus on the most important and at the same time clinically relevant topics. A better structure and shorter version would significantly improve the study and maintain the excitement of reading throughout the entire manuscript. The reviewer recommends that the aforementioned limitations be addressed before the study can be considered for publication in PLOS ONE.

See the detailed comments on each section below.

Title

Much better now. Adequate title.

Abstract

According to the submission guidelines (https://journals.plos.org/plosone/s/submission-guidelines), the abstract should not exceed 300 words. In the current version, the abstract has more than 400 words. Please shorten the abstract and focus on the most important information.

Background: Adequate.

Methods: Line 50-53: In order to shorten the abstract, the authors consider to delete this information from the abstract. However, this information is of utmost importance in the main manuscript.

Results: Adequate.

Conclusions: Despite the response of the authors, the conclusion is still too long. Try to shorten.

Introduction

The introduction is far too long. This was already mentioned by both reviewers in the first review. Try to shorten the introduction to no more than 500 words.

Line 69-86: The information in the first paragraph is well known. The authors should summarize the most important information regarding the submitted systematic review in 1-2 sentences or consider deleting the first paragraph.

Line 87-98: This paragraph is a good introduction into the topic of the systematic review.

Line 109-111: Consider to delete this part.

Material and Methods

The authors did a good job on revising the materials and methods section. No further comments.

Results

The authors may shorten the results section. Try to avoid repetitions between tables and written text.

Moreover, the authors did a good job on revising the results section. No more comments.

Discussion

The discussion section is still far too long. It is not intended to report all studies of the research topic and their results. Rather, a discussion / interpretation of the most important findings of the systematic review with already existing literature should be performed. Controversies should be discussed and clinically relevant results highlighted.

Conclusion

The conclusion section is still too long. The authors should shorten the conclusion and focus on the most significant and clinically relevant findings.

Line 480-486: The reviewer recommends to move this part to the discussion section.

Figures + Tables

No further comments.

References

No further comments.

7. PLOS authors have the option to publish the peer review history of their article (what does this mean?). If published, this will include your full peer review and any attached files.

Reviewer #1: No

Reviewer #2: No

---

## [Author Response · Author response to Decision Letter 1]

5 Dec 2022

Re-review of revised manuscript ‘’Effects of The Menstrual Cycle Phase on Anterior Cruciate Ligament Neuromuscular and Biomechanical Injury Risk Surrogates in Eumenorrheic and Naturally Menstruating Women: A Systematic Review and Recommendations for Future Research’’, Dos’Santos et al.

Reviewer: Alexandra Horvath, PONE-D-22-07730R1

Abstract: 

On line 59, please change (7-9 Modified Downs and Black Checklist score 7-9; and GRADE) so that it is understandable which score corresponds to which quality assessment/grading tool.

Response: Thank you for your comment. In a previous round of revisions, we were told to include this. Both of these tools use descriptors to classify the evidence as low or very low.

Line 65-66 – remove this sentence as it does not add additional value to the conclusion section.

Response: Thank you for your comment. This has been removed.

Methods:

In table I ‘’study design’’, please change experimental to clinical or human. 

Response: Thank you for your comment. This has been amended.

Who performed the literature search in PubMed, Medline (OVID), SPORTDiscus, and Web of Science databases? 

Response: Thank you for your comment. This has been added. Please see line 117.

Line 169, please add the reference to the guideline that was used. 

Response: Thank you for your comment. This has been added. Please see line 119.

Results

I would recommend that ‘’statistically’’ is removed from ‘’no statistically significant’’ on line 255. Furthermore, what does ‘’meaningful difference’’ refer to on line 255? 

Response: Thank you for your comment. This has been removed throughout the document. Please see line section 3. Meaningful is a term to denote inferences regarding the magnitude of differences, as suggested by Will Hopkins https://www.sportsci.org/jour/05/ambwgh.htm For instance, there may indeed be significant differences observed; however, the magnitude of the differences (i.e., effect size) may be trivial / small and thus not practically or clinically meaningful. Conversely, you may observe no significant differences; however, a practically or clinically meaningful effect size could be observed. As such, we would like to keep the term “meaningful” in the manuscript. 

Discussion

The sentence on line 305-308 is very long, I would suggest that the sentence is divided into two. 

Response: Thank you for your comment. This has been amended to two sentences.

Please remove reference to table on line 454. 

Response: Thank you for your comment. This has been removed.

Line 461 – please add which type of bias(es) the review was subjected to. 

Response: Thank you for your comment. This has been added. Please see line 354-356.

Conclusion

The conclusion section is far too long. It needs to be shortened. 

Response: Thank you for your comment. This has been reduced from 277 to 186 words

Reviewer #2: Dear authors,

The first revision substantially improved the quality of the study. However, the biggest limitation of the manuscript is that it is still far too long. In particular, the introduction, discussion, and conclusion sections need to be shortened. The authors should focus on the most important and at the same time clinically relevant topics. A better structure and shorter version would significantly improve the study and maintain the excitement of reading throughout the entire manuscript. The reviewer recommends that the aforementioned limitations be addressed before the study can be considered for publication in PLOS ONE.

See the detailed comments on each section below.

Title

Much better now. Adequate title.

Response: Thank you for your comment.

Abstract

According to the submission guidelines (https://journals.plos.org/plosone/s/submission-guidelines), the abstract should not exceed 300 words. In the current version, the abstract has more than 400 words. Please shorten the abstract and focus on the most important information.

Response: Thank you for your comment. This has been reduced from 402 to 300 words.

Background: Adequate.

Methods: Line 50-53: In order to shorten the abstract, the authors consider to delete this information from the abstract. However, this information is of utmost importance in the main manuscript. 

Response: Thank you for your comment and suggestion. This has been amended.

Results: Adequate.

Response: Thank you for your comment.

Conclusions: Despite the response of the authors, the conclusion is still too long. Try to shorten.

Response: Thank you for your comment. This has been reduced from 90 to 74 words.

Introduction

The introduction is far too long. This was already mentioned by both reviewers in the first review. Try to shorten the introduction to no more than 500 words.

Response: Thank you for your comment. We have amended the introduction, removing two main paragraphs and therefore have 4 short paragraphs. As such, the word count is now 691 words, having previously been 1329 words. We have done our best to remove all unnecessary information but feel the remaining content must remain in the introduction to provide an appropriate background and context. Overall, the word count is quite short for an introduction, and the total word count is 4102 words which I am sure you agree is low for a systematic review.

Line 69-86: The information in the first paragraph is well known. The authors should summarize the most important information regarding the submitted systematic review in 1-2 sentences or consider deleting the first paragraph.

Response: Thank you for your comment. We have deleted this first paragraph.

Line 87-98: This paragraph is a good introduction into the topic of the systematic review.

Response: Thank you for your comment. We have started from here. Please see line 61.

Line 109-111: Consider to delete this part.

Response: Thank you for your comment. We have deleted this.

Material and Methods

The authors did a good job on revising the materials and methods section. No further comments.

Response: Thank you for your comment.

Results

The authors may shorten the results section. Try to avoid repetitions between tables and written text.

Moreover, the authors did a good job on revising the results section. No more comments.

Response: Thank you for your comment. We have removed ~220 words, reducing the word count from 886 words to 664 words.

Discussion

The discussion section is still far too long. It is not intended to report all studies of the research topic and their results. Rather, a discussion / interpretation of the most important findings of the systematic review with already existing literature should be performed. Controversies should be discussed and clinically relevant results highlighted.

Response: Thank you for your comment. We have removed ~700 words from the discussion, reducing the word count 2142 words to 1375 words, leaving the most clinically relevant results and applications.

Conclusion

The conclusion section is still too long. The authors should shorten the conclusion and focus on the most significant and clinically relevant findings. Line 480-486: The reviewer recommends to move this part to the discussion section.

Response: Thank you for your comment. This has been reduced from 277 to 185 words.

---

## [Decision Letter · Decision Letter 2]

10 Jan 2023

Effects of The Menstrual Cycle Phase on Anterior Cruciate Ligament Neuromuscular and Biomechanical Injury Risk Surrogates in Eumenorrheic and Naturally Menstruating Women: A Systematic Review

PONE-D-22-07730R2

Dear Dr. Dos'Santos,

We’re pleased to inform you that your manuscript has been judged scientifically suitable for publication and will be formally accepted for publication once it meets all outstanding technical requirements.

Kind regards,

Javier Peña, Ph.D.

Academic Editor

PLOS ONE

Reviewers' comments:

Reviewer's Responses to Questions

**Comments to the Author**

1. If the authors have adequately addressed your comments raised in a previous round of review and you feel that this manuscript is now acceptable for publication, you may indicate that here to bypass the “Comments to the Author” section, enter your conflict of interest statement in the “Confidential to Editor” section, and submit your "Accept" recommendation.

Reviewer #1: All comments have been addressed

Reviewer #2: All comments have been addressed

2. Is the manuscript technically sound, and do the data support the conclusions?

Reviewer #1: Yes

Reviewer #2: Yes

3. Has the statistical analysis been performed appropriately and rigorously? 

Reviewer #1: N/A

Reviewer #2: Yes

4. Have the authors made all data underlying the findings in their manuscript fully available?

Reviewer #1: Yes

Reviewer #2: Yes

5. Is the manuscript presented in an intelligible fashion and written in standard English?

Reviewer #1: Yes

Reviewer #2: Yes

6. Review Comments to the Author

Reviewer #1: Re-revision of revised manuscript ‘’Effects of The Menstrual Cycle Phase on Anterior Cruciate Ligament Neuromuscular and Biomechanical Injury Risk Surrogates in Eumenorrheic and Naturally Menstruating Women: A Systematic Review and Recommendations for Future Research’’, Dos’Santos et al.

I have no further comments to add.

Reviewer #2: Dear authors,

The second revision substantially improved the quality of the study. The systematic review is well done and interesting. The reviewer recommends that the manuscript be accepted for publication in PLOS ONE in its current form.

Title

Adequate.

Abstract

The abstract in its revised version is good.

Introduction

The authors did a good job in revising the introduction section. The authors make a good introduction into the topic of the systematic review! Adequate objective. Adequate hypothesis. No further comments.

Material and Methods

No further comments.

Results

The authors did a good job in revising the results section. No further comments.

Discussion

The authors did a good job in revising the discussion section. No further comments.

Conclusion

No further comments.

Figures + Tables

No further comments.

References

No further comments.

7. PLOS authors have the option to publish the peer review history of their article (what does this mean?). If published, this will include your full peer review and any attached files.

Reviewer #1: No

Reviewer #2: No

---

## [Editor Report · Acceptance letter]

17 Jan 2023

PONE-D-22-07730R2 

Effects of The Menstrual Cycle Phase on Anterior Cruciate Ligament Neuromuscular and Biomechanical Injury Risk Surrogates in Eumenorrheic and Naturally Menstruating Women: A Systematic Review 

Dear Dr. Dos'Santos:

I'm pleased to inform you that your manuscript has been deemed suitable for publication in PLOS ONE. Congratulations! Your manuscript is now with our production department. 

Kind regards, 

on behalf of

Dr. Javier Peña 

Academic Editor

PLOS ONE